# Neural JNK3 regulates blood flow recovery after hindlimb ischemia in mice via an Egr1/Creb1 axis

Shashi Kant [1,2], Siobhan M. Craige[1,3], Kai Chen[1,4], Michaella M. Reif[1,5], Heather Learnard [1], Mark Kelly [1], Amada D. Caliz[1,2], Khanh-Van Tran[1], Kasmir Ramo[6], Owen M. Peters[7,8], Marc Freeman[7,9], Roger J. Davis [6,10] & John F. Keaney Jr.[1,2]

Diseases related to impaired blood flow such as peripheral artery disease (PAD) impact nearly 10 million people in the United States alone, yet patients with clinical manifestations of PAD (e.g., claudication and limb ischemia) have limited treatment options. In ischemic tissues, stress kinases such as c-Jun N-terminal kinases (JNKs), are activated. Here, we show that inhibition of the JNK3 (Mapk10) in the neural compartment strikingly potentiates blood flow recovery from mouse hindlimb ischemia. JNK3 deficiency leads to upregulation of growth factors such as *Vegfa*, *Pdgfb*, *Pgf*, *Hbegf* and *Tgfb3* in ischemic muscle by activation of the transcription factors Egr1/Creb1. JNK3 acts through Forkhead box O3 (Foxo3a) to suppress the activity of Egr1/Creb1 transcription regulators in vitro. In JNK3-deficient cells, Foxo3a is suppressed which leads to Egr1/Creb1 activation and upregulation of downstream growth factors. Collectively, these data suggest that the JNK3-Foxo3a-Egr1/Creb1 axis coordinates the vascular remodeling response in peripheral ischemia.

---

[1] Division of Cardiovascular Medicine, Department of Medicine, University of Massachusetts Medical School, Worcester, MA 01655, USA. [2] Division of Cardiovascular Medicine, Department of Medicine, Brigham Women's Hospital, Harvard Medical School, Boston, MA 02115, USA. [3] Human Nutrition, Foods, and Exercise, Virginia Tech, VA 24061, USA. [4] Department of Medicine, University of Connecticut Health Center, Farmington, CT 06030, USA. [5] University of Maryland School of Medicine, Baltimore, MD 21201, USA. [6] Program in Molecular Medicine, University of Massachusetts Medical School, Worcester, MA 01655, USA. [7] Department of Neurobiology, University of Massachusetts Medical School, Worcester, MA 01655, USA. [8] Cardiff University, School of Biosciences, Cardiff CF10 3AX, UK. [9] The Vollum Institute Oregon Health & Science University, Portland, OR, USA. [10] Howard Hughes Medical Institute, Worcester, MA 01605, USA. Correspondence and requests for materials should be addressed to S.K. (email: skant1@bwh.harvard.edu) or to J.F.K.Jr. (email: jfkeaney@bwh.harvard.edu)

Peripheral arterial disease (PAD) refers to a diverse group of disorders that result in impaired blood flow leading to critical limb ischemia (CLI) and pain (claudication) with exercise or prolonged activity, largely due to inadequate arteriogenesis (formation of collateral vessels) and poor angiogenesis (formation of new blood vessels)[1]. This condition impacts up to 14% of the population, aged 70 and older, and is particularly prevalent in those patients with type 2 diabetes mellitus[2]. More than 200,000 individuals undergo lower-limb amputation every year because of peripheral vascular diseases[1,2]. Although nearly 10 million individuals in the United States suffer from PAD[3,4], there is a paucity of effective treatment options largely due to an incomplete understanding of the molecular responses to ischemia.

Among the stress-induced pathways activated during ischemia, those relevant to angiogenesis are the c-Jun N-terminal kinase (JNK) family of protein kinases[5,6]. Three separate genes encode JNK1 (*Mapk8*), JNK2 (*Mapk9*), and JNK3 (*Mapk10*), and alternative splicing can produce 10 different protein sequences with significant homology[7,8]. JNK1 and JNK2 are expressed ubiquitously, but JNK3 expression is mainly confined to neurons, pancreas, testis, and the heart[7,8]. Alterations in JNK1 and JNK2 signaling are now well-established causes of metabolic diseases such as diabetes and chronic inflammation[9–14], which have been further linked to the onset and progression of cardiovascular disease. A recent study implicated JNK1 and JNK2 in angiogenesis during development[15] but the role of JNK3 in the regulation of neovascularization is not known.

Early growth response (EGR) genes belongs to the zinc finger transcription factor (TF) superfamily. EGR has four family members, denoted EGR1, EGR2, EGR3, and EGR4[16]. As the name suggests, EGR genes are amongst the first pool of genes activated by different physiological stimuli[16,17]. Once stimulated, EGRs regulate the expression of a vast variety of genes including growth factors such as vascular endothelial growth factor (VEGF)[17]. The role of EGR1 in vascular homeostasis has been implicated previously[18]. Though EGR1 is not expressed in the neuronal compartment during development, it is expressed in adult neurons and controls neuronal activity[16].

The cAMP response element binding protein 1 (CREB1) is a transcription factor belonging to the bZIP superfamily[19]. CREB1, along with its two closely related bZIP superfamily members, activating transcription factor 1 (ATF1) and cAMP response element modulator (CRAM) transcription factor, form the Creb family[19]. Creb family members contain both a leucine zipper domain that enables dimerization and a C-terminal basic domain that facilitates DNA binding. Creb1 family members can form homo and hetro-dimmers which regulate transcription of downstream genes by binding to cAMP response element (CRE)[19]. Creb1 can be activated by many diverse stimuli and its activation occurs mainly by phosphorylation at its serine 133[20] phosphorylation site. Creb1 plays an important role in neural development and is required for neuronal survival[19]. Mice lacking Creb1 show apoptosis, axonal growth defects, and degeneration of peripheral neurons[21]. It is also known that Creb1 can regulate expression of different pro-angiogenic factors[22,23].

The forkhead box (FOX) transcription factors play a pivotal role in development, proliferation, differentiation, and stress resistance[24]. The forkhead box family of transcription factors contains a group of more than 100 proteins with the 'O' subgroup consisting of four members (FOXO1, FOXO3, FOXO4, and FOXO6)[24]. Nuclear localization of FOXO proteins is regulated through their phosphorylation. Phosphorylation by AKT/PKB of the Foxo3a[25] protein at Thr32, Ser253, and Ser315 facilitates Foxo3a binding to the 14-3-3 proteins[25,26]. This AKT-induced binding of phosphorylated Foxo protein to 14-3-3 is responsible for Foxo cytoplasmic localization[27,28]. The MAPK family member, c-JUN N-terminal kinase (JNK) is able to phosphorylate 14-3-3 at a Ser184/186 site[29] which leads to Foxo protein release from 14-3-3 with subsequent localization to the nucleus[28,29], where Foxo can suppress transcription of various genes including EGR1[30,31]. In addition, *Foxo3a* deficiency leads to enhanced postnatal vessel formation and maturation[32].

Here we explore the role of JNK3 in blood flow recovery after hindlimb ischemia via regulation of Foxo3a. We show that JNK3-deficiency reduces the phosphorylation of 14-3-3 on Ser184/186 which in turn decreases the nuclear localization of Foxo3a. Furthermore, JNK3 deficiency increases the activation of transcription factors Egr1 and Creb1. During hypoxia induction, Egr1 interaction with Creb1 substancially increases which leads to the subsequent upregulation of growth factors to facilitate blood flow recovery after hindlimb ischemia.

## Results

### JNK3 expression increases after ischemia in human and mouse.

In patients with critical limb ischemia undergoing lower extremity amputations, we found that phosphorylation of JNK proteins in the nerve and muscle tissues was enhanced in areas of compromised blood supply compared with normally perfused areas (Supplementary Fig. 1a, b). Furthermore, we found that JNK3 expression was upregulated in areas with ischemia, compared with areas with intact blood supply (Fig. 1a). To study how JNK3 may impact critical ischemia further, we utilized a hindlimb ischemia (HLI) model of PAD, where ischemia is induced by ligating the femoral artery and blood flow recovery is quantified by laser Doppler perfusion imaging (LDPI) over time. We observed a significant increase in *Mapk10* mRNA expression (JNK3) in the gastrocnemius muscle of wild-type mice with HLI (Fig. 1b). As the gastrocnemius is a mixed tissue and *Mapk10* is primarily expresses in neurons[33], we isolated peripheral nerves to investigate nerve-specific JNK expression with ischemia. There was a marked increase in *Mapk10* RNA expression in peripheral nerves with ischemia (Supplementary Fig. 1c) without any change in other members of the JNK family (*Mapk8* and *Mapk9*) that are expressed in neurons. These data suggested that ischemia regulates *Mapk10* expression in the peripheral nerves that may influence the blood flow recovery process.

### JNK3-deficiency enhances the blood flow recovery.

To determine the role of JNK3 in ischemia-induced blood flow recovery we tested the HLI model in wild-type (WT) and whole-body *Mapk10*−/− mice. Unlike *Mapk8*− and *Mapk9*−-deficient mice that exhibited suppressed blood flow recovery[15], mice lacking *Mapk10* had significantly enhanced blood flow recovery in response to HLI compared with littermate WT controls (Fig. 1c, d).

Since multiple cell types, including endothelium, smooth muscle, and neurons, are involved in orchestrating the blood flow recovery after ischemia[34], we investigated *Mapk10* expression in these tissues (Supplementary Figs. 1d–f, 2a–c). The endothelium plays a vital role in both ischemia-induced collateral artery remodeling and neovascularization[35,36]. Primary endothelium from mouse lung and skeletal muscle did not exhibit any *Mapk10* expression (Supplementary Fig. 1e). Since we detected *Mapk10* mRNA in the C2C12 mouse skeletal muscle cell line, we investigated this tissue using a skeletal muscle-specific cre recombinase (human alpha-skeletal actin; HSA cre[37]) to specifically delete the *Mapk10* gene in mouse skeletal muscle (Supplementary Fig. 2a, b). Skeletal muscle-deficient *Mapk10* mice exhibited no difference in blood flow recovery compared with controls after HLI (Fig. 2b). As *Mapk10* is known to be

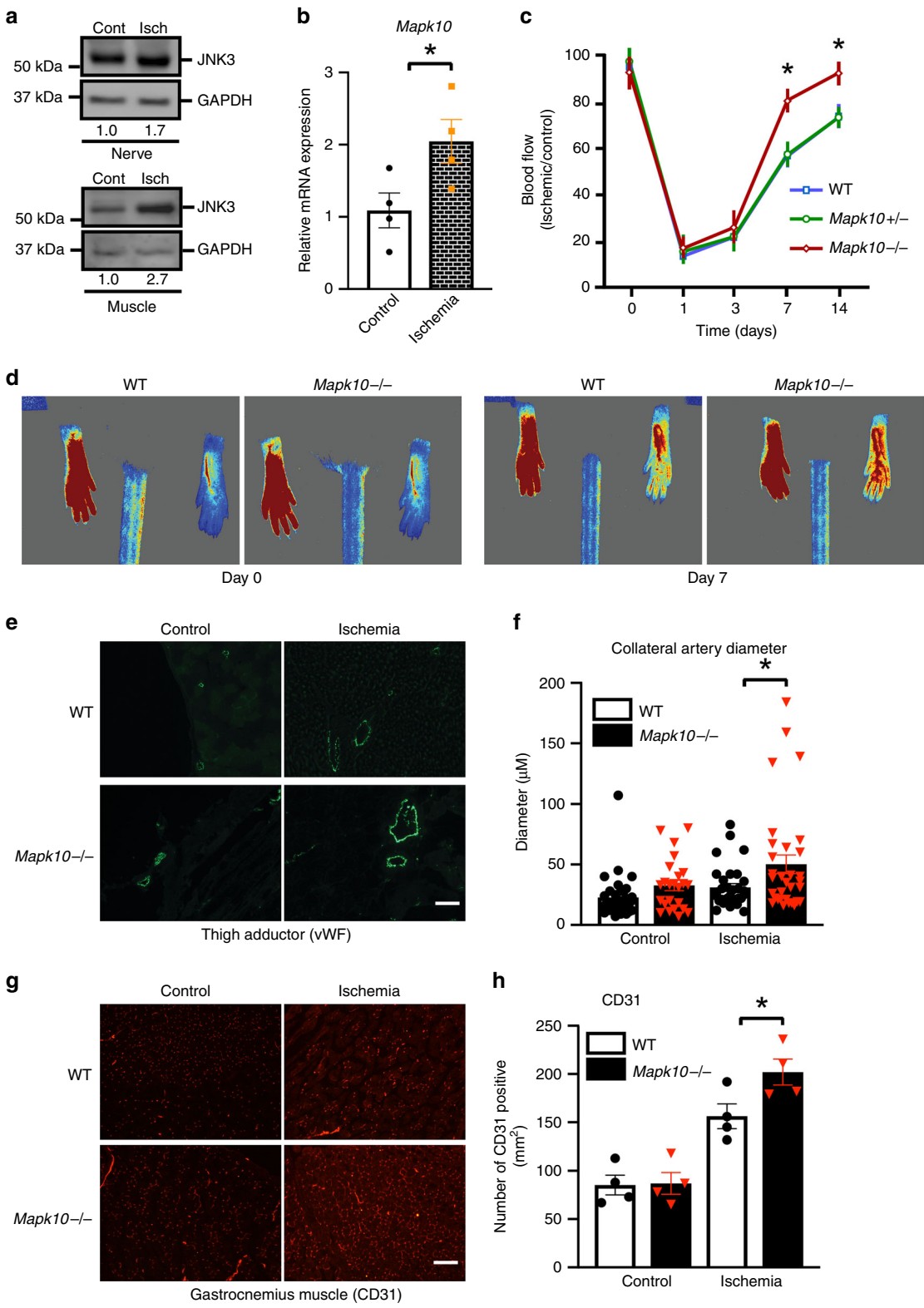

highly expressed in neural tissue[8], we used a nestin promotor-driven cre (Nes cre)[38] recombinase to specifically delete the *Mapk10* gene from both central and peripheral neurons (Supplementary Fig. 2a). Similar to whole-body *Mapk10*-deficient animals, mice lacking neural *Mapk10* had accelerated blood flow recovery in response to HLI as compared with controls (Fig. 2a, c). Based on these observations, we concluded that neural

*Mapk10* plays a key role in blood flow recovery in peripheral ischemia.

Ischemic blood flow recovery involves many processes, including the restructuring of the collateral arteries. To assess the impact of Mapk10 on collateral artery remodeling after HLI, we examined arterial markers, collateral artery diameter and circumference post HLI in WT and *Mapk10*-deficient mice[39].

**Fig. 1** Loss of JNK3 (Mapk10) promotes blood flow recovery after hindlimb ischemia (HLI). JNK3 protein expression from human peripheral nerves and muscle (**a**) as well as mRNA expression from mouse gastrocnemius muscle (**b**) were measured by western blot or qRT-PCR after tissues were harvested from either control or ischemic areas of peripheral limbs (n = 4 in each group). **c** Time course of blood flow recovery by laser Doppler imaging in HLI of mice with the indicated genotype (WT, n = 4; Mapk10[+/−], n = 5; Mapk10[−/−], n = 4). **d** Representative figures of blood flow measurements by laser Doppler ilmaging in hindlimb ischemia model at day 7 post HLI of control and Mapk10[−/−] mice. **e** Expression of von Willebrand factor (vWF) in mouse thigh adductor muscle on day 21 after femoral artery ligation. **f** Quantification of diameter of collateral vessels, vWF, in mouse thigh adductor muscle on day 21 after femoral artery ligation (WT control, n = 32; Mapk10[−/−] control, n = 24; WT ischemia, n = 31; Mapk10[−/−] ischemia, n = 32). **g** Cluster of differentiation 31 (CD31) also known as platelet endothelial cell adhesion molecule (PECAM-1) expression in mouse gastrocnemius muscle on day 21 after femoral artery ligation. **h** CD31 quantification, in mouse gastrocnemius muscle on day 21 after femoral artery ligation (n = 4 in each group). *P < 0.05 vs. indicated comparison by Student's t test. The data are mean ± SEM. Scale bar, 60 μm. Source data are provided as a Source Data file

Collateral artery diameters and circumference of whole-body Mapk10 knockout as well as neural Mapk10-deficient mice were significantly higher than control littermates after hindlimb ischemia (Figs. 1e, f, 2d, e, Supplementary Fig. 3a, b). Furthermore, we also observed that a few known markers for collateral remodeling[36,40] were upregulated in both whole-body Mapk10 knockout and neural Mapk10-deficient mice hind limb compared with control littermates (Supplementary Fig. 3c, Fig. 2f). Using endothelial CD31 immunostaining as an index of capillary formation, we found enhanced capillary formation in the gastrocnemius muscle of Mapk10[−/−] mice compared with WT after HLI (Fig. 1g, h).

As neural knockout of JNK3 facilitates an improved ischemic response, we asked whether this was due to JNK3 protection of neurons from ischemic insult. Sterile alpha and TIR motif containing 1 (Sarm1) protein was shown to play an important role in the regulation of post-injury peripheral neuron survival wherein Sarm1 knockout prevents axonal self-destruction after injury[41]. We tested our theory of blood flow recovery using Sarm1 knockout mice. However, we did not see any difference in blood flow recovery between Sarm1-deficient and control mice after hindlimb ischemia (Supplementary Fig. 4a). Furthermore, we conducted sciatic nerve axotomy on both WT and Mapk10-null mice, and unlike what has been reported in Sarm1 deficiency[41], Mapk10-deficient mice exhibited no axonal protection after sciatic nerve axotomy (Supplementary Fig. 4b). These data indicate there is no axonal preservation after injury with Mapk10 deletion and that the improvement in ischemic blood flow recovery in Mapk10-deficient mice occurs through other mechanisms.

Immune cells have also been implicated in blood flow recovery after HLI[42,43]. Therefore, we investigated the expression of macrophage and T-cell markers in Mapk10-deficient mice after HLI. We saw no differences in the expression of immune cell marker genes such as Cd68, Cd4, and Cd3 between Mapk10-deficient and control mice (Supplementary Fig. 5). Thus, immune cells are not likely to play a central role in the observed enhancement of ischemic blood flow recovery in Mapk10-deficient mice.

**JNK3 suppresses the expression of growth factors**. In order to ascertain the molecular mechanisms responsible for the accelerated blood flow recovery in Mapk10-deficient mice, a microarray analysis of gastrocnemius muscle from these mice was performed after HLI. We observed a significant upregulation of several growth factors[44–48] known to improve blood flow recovery in the Mapk10[−/−] muscle compared with WT (Figs. 3a–d). Specifically, qRT-PCR confirmed upregulation of Vegfa, Pdgfb, Pgf, Hbegf, and Tgfb3 (Fig. 3b), with a trend in neuropeptide production[49] and no material impact on mitochondrial biogenesis[50] markers (Fig. 3c, d). Next, we examined the expression of Vegfa, Pdgfb, Pgf, Hbegf, and Tgfb3 in neuron-specific Mapk10 deleted Nes[+/Cre];Mapk10[f/f] mice. Similar to whole-body Mapk10[−/−]

mice, neural Mapk10-deficient mice also have an upregulation of Vegfa, Pdgfb, Pgf, Hbegf, and Tgfb3 gene expression in gastrocnemius muscle (Fig. 3e). Immunoblot analysis confirmed increased protein expression of PDGF-B (Fig. 3f) and VEGF-A (Fig. 3g) in Mapk10-deficient gastrocnemius muscle after ischemia. We observed a similar upregulation of VEGF-A protein in gastrocnemius muscle of neuron-specific Mapk10-deficient mice (Fig. 3h). Taken together, these results demonstrate that neural Mapk10 deletion promotes pathways involved in improved blood flow recovery after ischemia.

As Mapk10 is highly expressed in neural tissue[8], we examined the growth factor profile from peripheral nerves in our Mapk10[−/−] mice. We found that the pro-angiogenic genes of Vegfa, Pdgfb, and Hbegf were indeed upregulated in Mapk10[−/−] peripheral nerves as compared with those from WT littermates (Fig. 4a). We confirmed the association between neural Mapk10 expression and pro-angiogenic gene expression using the neuroblastoma cell line Neuro-2a (N2a). In this cell line, Mapk10 knockdown increased expression of Vegfa, Pdgfb, and Hbegf in hypoxic conditions (Fig. 4b). Unlike the N2a cell line, C2C12 muscle cells exhibited no difference in expression of pro-angiogenic factors after Mapk10 knockdown (Supplementary Fig. 6).

In order to confirm that Mapk10 loss from the peripheral nerves leads to increased growth factor expression in vivo, we performed Mapk10 gene silencing utilizing a gene painting technique adapted from the heart[51]. We observed that inhibition of Mapk10 in peripheral nerves leads to an increase in Pdgfb and Vegfa expression (Fig. 4c), suggesting that peripheral nerve-specific Mapk10 inhibits the expression of growth factors in neurons.

**EGR1 interacts with Creb1 and controls the growth factors**. Two known transcriptional regulators of growth factors, Egr1 and Creb1, were significantly upregulated in Mapk10[−/−] mice in our microarray assay and qRT-PCR (Figs. 3a, 4d). Both, Egr1 and Creb1 have been implicated in several pro-angiogenic pathways and blood flow recovery after HLI[18,50,52,53]. To determine whether Mapk10 mediates Egr1 protein expression in neural cell lines, we utilized immunostaining for Egr1 after Mapk10 knockdown N2a cells. We found that Egr1 expression was increased in Mapk10 diffficient cells when compared with control (Figs. 4a–c, 5a). Furthermore, to determine the nuclear localization of Egr1, we conducted a nuclear fractionation assay following siRNA treatment of the N2a cell line and discovered Egr1 was more abundant in the nuclear fraction of Mapk10 siRNA-treated cells than those treated with control siRNA (Fig. 5b). Using an Egr1-luciferase reporter assay, we found that Egr1 transcriptional activity was increased with Mapk10 siRNA treatment compared with control siRNA (Fig. 5c). Thus, JNK3 regulates both expression and activity of the Egr1 protein.

To determine the importance of Egr1 in neuronal cells we utilized both loss- and gain-of-function strategies. Egr1 siRNA treatment of N2a cells produced downregulation of the growth

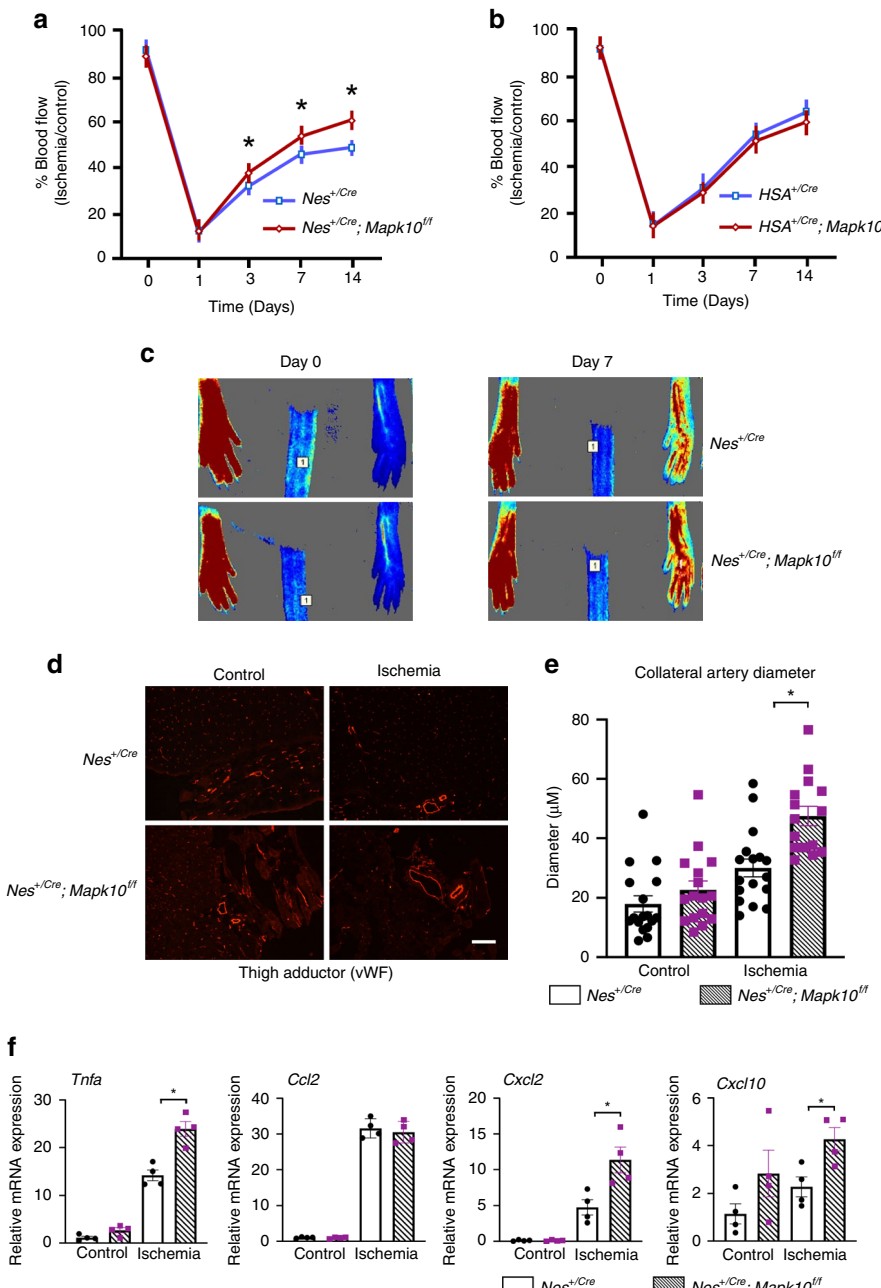

**Fig. 2** Neural JNK3 plays an important role during blood flow recovery after hindlimb ischemia (HLI). Time course of blood flow recovery by laser speckle contrast imaging of neural-specific nestin cre (**a**) and muscle-specific HSA cre (**b**) *Mapk10−/−* mice (*Nes+/Cre*, n = 9; *Nes+/Cre Mapk10f/f*, n = 8; *HSA+/Cre*, n = 8; *HSA+/Cre Mapk10f/f*, n = 8). **c** Representative figures of blood flow measurements in the HLI model model on day 7 post HLI in *Nes+/Cre* and *Nes +/Cre; Mapk10f/f* mice. **d** Expression of von Willebrand factor (vWF) in mouse thigh adductor muscle on day 21 after femoral artery ligation. **e** Quantification of the diameter of collateral vessels, vWF, in mouse thigh adductor (TA) muscle on day 21 after femoral artery ligation (*Nes+/Cre* control, n = 17; *Nes+/Cre Mapk10f/f*, n = 16; *Nes+/Cre* ischemia, n = 17; *Nes+/Cre Mapk10f/f* ischemia, n = 15). **f** qRT-PCR was performed for different genes related to collateral artery remodeling on thigh adductor muscle from the control and ischemic legs of WT and *Nes+/Cre; Mapk10f/f* mice 3 days post femoral artery ligation (n = 4 in each group). All the experiments have been repeated 3–6 times. Statistically significant differences between groups are indicated (*P < 0.05 by Student's t test). The data are mean ± SEM. Scale bar, 50 μm. Source data are provided as a Source Data file

factors *Vegfa*, *Pgf*, and *Hbegf* (Fig. 5g). Conversely, overexpression of Egr1 in N2a cells led to upregulation of *Pdgfb* and *Pgf* (Fig. 5h). Collectively, the data indicates that Egr1 regulates the expression of multiple growth factors involved in blood flow recovery from ischemia.

In order to determine the role of Egr1 in ischemia-induced blood flow recovery in vivo, we over-expressed *Egr1* in the hind limb of mice by using an *Erg1* expressing adenoviral vector.

Similar to the phenotype of *Mapk10*-deficient mice, *Egr1* gain-of-function mice exhibited enhanced blood flow recovery compared with control mice following HLI (Fig. 5d, f). These results strongly suggest that increased *Egr1* expression and activity in *Mapk10−/−* mice can promote enhanced blood flow recovery.

Egr1 and Creb1 interact with Creb1 binding protein (CBP) and regulate the transcription of multiple genes[54]. To determine if enhanced Egr1-mediated gene expression might involve Creb1,

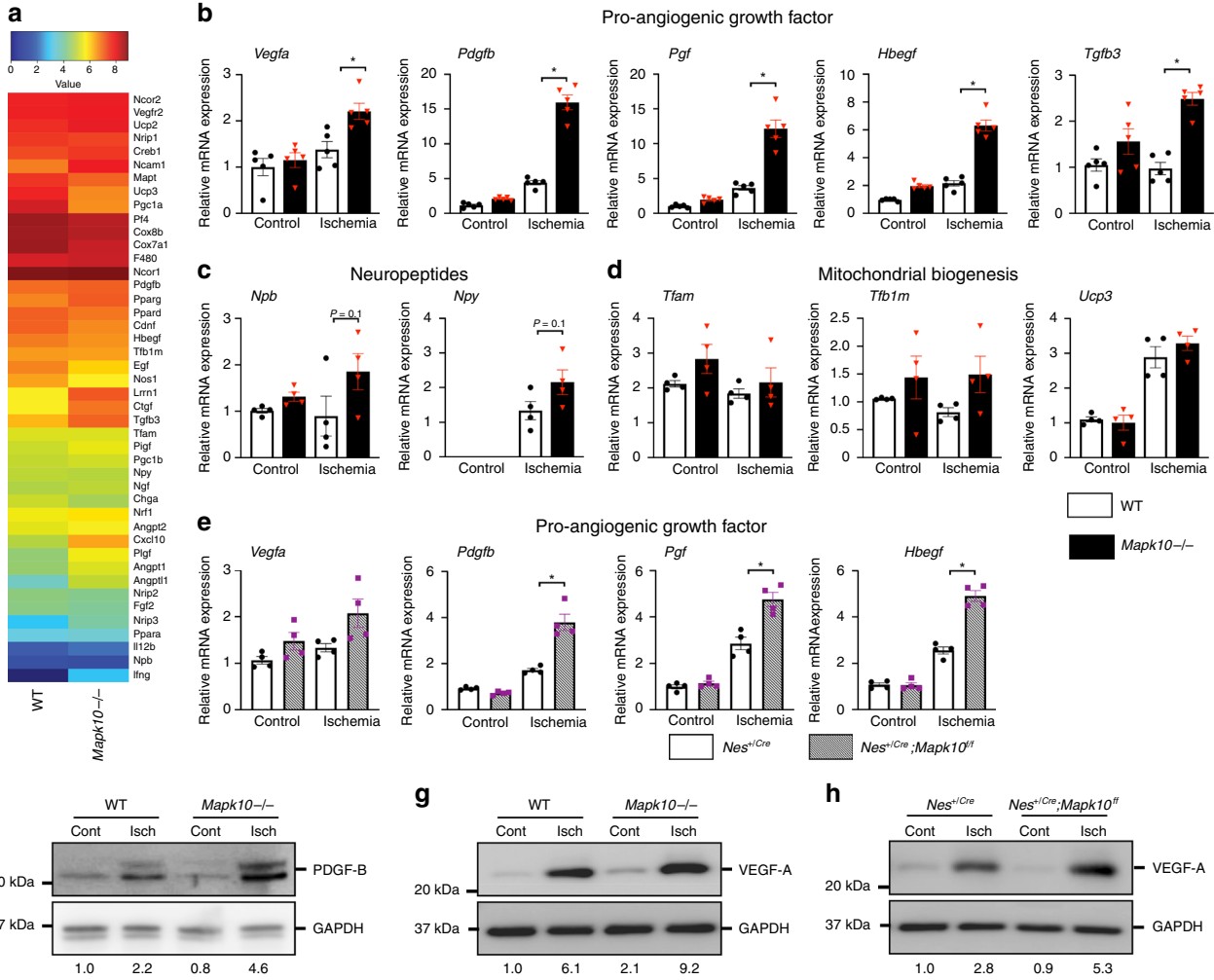

**Fig. 3** Suppression of JNK3 activates pro-angiogenic genes in gastrocnemius muscle during ischemia. **a** Microarray was performed on pooled (three mice) gastrocnemius muscle three days post femoral artery ligation. **b–d** qRT-PCR was performed for different blood flow recovery related genes on gastrocnemius muscle from control and ischemic legs of WT and *Mapk10*$^{-/-}$ mice 3 days post femoral artery ligation ($n = 4$ in each group). **e** qRT-PCR was performed for growth factor related genes on gastrocnemius muscle from control and ischemic legs of Nes$^{+/Cre}$ and Nes$^{+/Cre}$; *Mapk10*$^{f/f}$ mice 3 days post femoral artery ligation ($n = 4$ in each group). **f–h** Gastrocnemius muscle was isolated and immunoblots were performed as described. All the experiments have been repeated 3–6 times. Statistically significant differences between groups are indicated (*$P < 0.05$ by Student's *t* test). The data are mean ± SEM. Source data are provided as a Source Data file

we exposed N2a cells to hypoxia and conducted an immunoprecipitation assay of Creb1 and found that the Egr1 interaction with Creb1 is substantially increased after hypoxia (Fig. 5e). These results suggest ischemia-induced upregulation of growth factor genes may involve Egr1/Creb1 binding.

**Creb1 plays an important role in blood flow recovery.** Creb1 activation is largely regulated by phosphorylation of its serine residue 133[19,20]. We observed enhanced Creb1 Ser133 phosphorylation in ischemic *Mapk10*$^{-/-}$ mouse tissue compared with wild-type animals (Fig. 6a). Creb1 also upregulates its own transcription[54], and consistent with this notion we observed increased Creb1 expression at both the mRNA and protein levels with ischemia in *Mapk10*$^{-/-}$ vs. control mice (Figs. 4d, 6a). These findings were further confirmed in N2a cells after *Mapk10* siRNA treatment (Fig. 6b).

We then examined the effect of JNK3 on Creb1 binding and transcriptional activity. We transfected a Creb1 response element (CRE)[19] into N2a cells and observed that suppression of *Mapk10* significantly increases Creb1 activity (Fig. 6c), suggesting that JNK3 suppression increases Creb1 activation. Next, we used

siRNA against *Creb1* in N2a cells and observed that growth factors such as *Pgf* and *Tgfb3* were downregulated with Creb1 silencing (Fig. 6f). In contrast, overexpression of Creb1 in N2a cells increased expression of *Pdgfb*, *Pgf*, and *Tgfb3* (Fig. 6g). Collectively, these data suggest that Creb1 regulates expression of pro-angiogenic growth factors in neural cells.

In order to determine the specific role of Creb1 in ischemia-induced blood flow recovery in vivo we over-expressed *Creb1* in the hind limb using adenovirus. Similar to the response in *Mapk10*$^{-/-}$ mice and Egr1 gain-of-function phenotype, *Creb1* gain-of-function accelerated blood flow recovery compared with control mice following HLI (Fig. 6d, e). These data strongly implicate that increased Egr1 and Creb1 complex formation and transcriptional activity play an important role in the *Mapk10*$^{-/-}$ phenotype of enhanced blood flow recovery.

**FOXO regulates Egr1 and Creb1 expression and activity.** Similar to the *Mapk10* knockout phenotype, forkhead box O 3a (Foxo3a) deficiency in mice is known to increase blood flow recovery after hindlimb ischemia[32]. The Foxo family of transcription factors are regulated, in part, by phosphorylation of

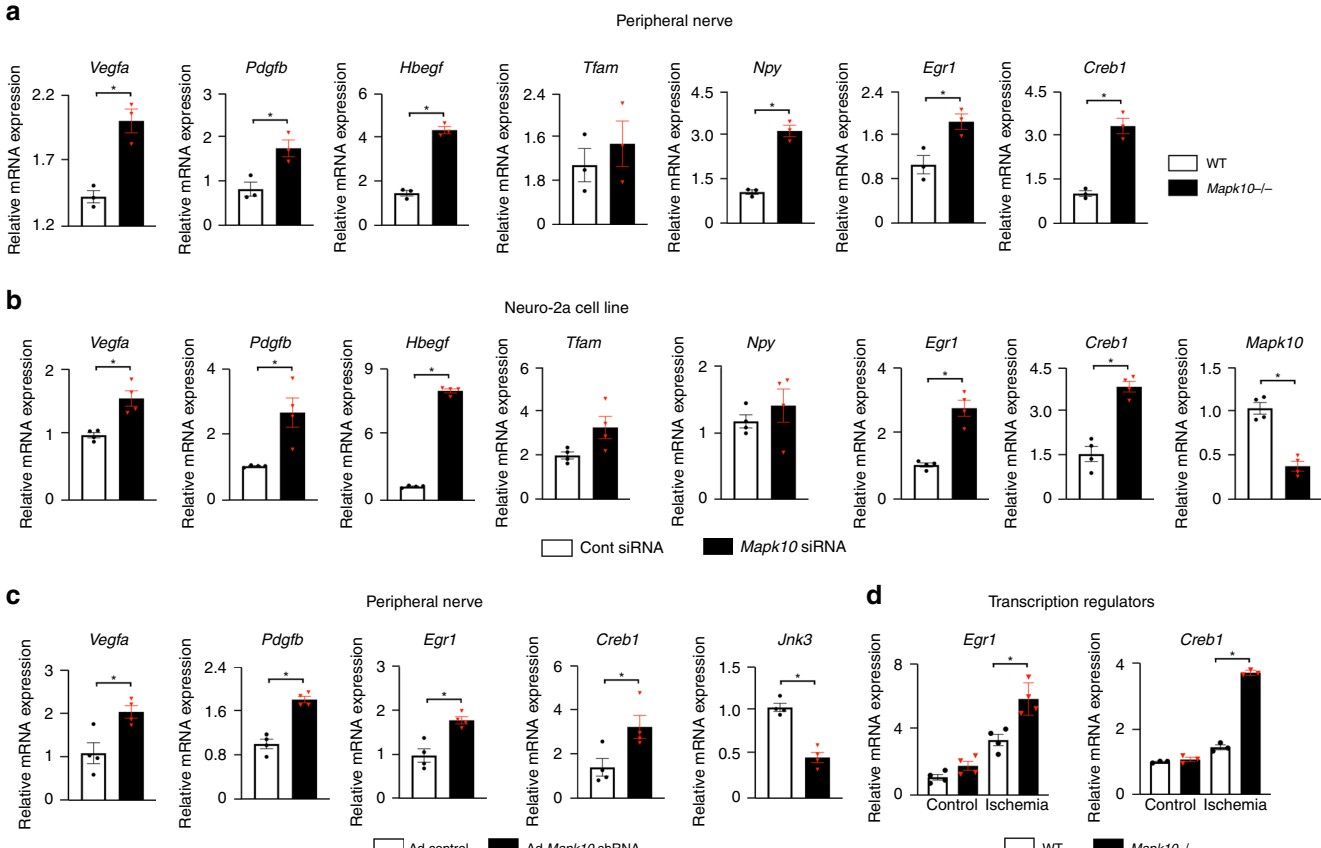

**Fig. 4** JNK3 regulates pro-angiogenic factors expression in femoral nerve and Neuro-2a (N2a) cell lines. **a** Femoral nerves were isolated on day 3 after femoral artery ligation and qRT-PCR was performed for different blood flow recovery related genes ($n = 3$ in each group). **b** Following 90 min of hypoxia, RNA was isolated from control and JNK3 knockdown N2a cells and qRT-PCR was performed for genes as indicated ($n = 4$ in each group). **c** Femoral nerves were isolated from control and ischemic legs of mice after 48 h of gene painting with adenovirus expressing *Mapk10* shRNA and qRT-PCR was performed for genes as indicated ($n = 4$ in each group). **d** Gastrocnemius muscle was isolated from control and ischemic legs of mice on day 3 after femoral artery ligation and qRT-PCR was performed as indicated. Statistically significant differences between groups are indicated (*$P < 0.05$ by Student's $t$ test). The data are mean ± SEM. Source data are provided as a Source Data file

multiple sites. Depending on phosphorylation site, Foxo's either stay in cytoplasm or translocates to the nucleus to modulate gene expression. The specific site of phosphorylation acts to either activate or suppress Foxo translocation to the nucleus that mediates Foxo-dependent downstream gene expression. Foxo3a phosphorylation by AKT/PKB[25,26] at Thr32 and Ser253 facilitates its binding to the 14-3-3 proteins, thereby sequestering Foxo3a in the cytoplasm[27,28]. Thus, we investigated Foxo3a phosphorylation at Thr32 and Ser253 in *Mapk10*-deficient cells by immunoblotting and found that Foxo3a phosphorylation at Thr32 and Ser253 were substantially increased in *Mapk10*-supressed cells compared with their wild-type counterparts (Fig. 7a). It is also known that 14-3-3 can be directly phosphorylated by JNK family members at Ser184/186 resulting in the release of Foxo family members from 14-3-3 to facilitate their nuclear translocation[28,29,55]. Thus, we probed 14-3-3 phosphorylation at Ser184/186 in *Mapk10*-shRNA-treated cells by immunostaining and found a marked decrease in 14-3-3 phosphorylation at Ser184/186 in *Mapk10*-supressed N2a cells (Fig. 7b, c). We next examined the role of Mapk10 on Foxo3a shuttling to the nucleus. We found that Mapk10 is required for Foxo3a nuclear accumulation as *Mapk10* shRNA-treated cells demonstrated decreased Foxo3a accumulation into the nucleus (Fig. 7d). Since another Foxo member, Foxo1, has been shown to suppress Egr1[31] activity and bind to Creb1[56], we examined the impact of Foxo3a loss-of-function on Creb1 and Egr1 expression and activity in the N2a cell line. Cells

treated with *Foxo3a* siRNA exhibited enhanced hypoxia-induced expression of both, Creb1 and Egr1 (Fig. 7e) compared with controls. Consistent with these data, Foxo3a supression yielded increased Creb1 and Egr1 transcriptional activity, mimicking the effect seen with the JNK3 inhibition (Fig. 7f, g).

Collectively, these data strongly suggest that JNK3 inhibits growth factor gene expression in ischemia via direct phosphorylation of the 14-3-3 protein at Ser184/186 which, in turn, releases the Foxo3a protein. Foxo3a then translocates to the nucleus where it mediates suppression of Egr1/Creb1, the transcriptional regulators of multiple pro-angiogenic growth factors, mitigating recovery from ischemia. Therefore, in the absence of JNK3, Foxo3a is trapped in the cytoplasm and Egr1/Creb1 promote uninhibited transcription of multiple growth factors that enhance blood flow recovery from ischemia (Fig. 7h).

## Discussion

Tissue ischemia, such as that seen in peripheral vascular disease is a major health burden. Here we have identified a molecular target, JNK3, that dramatically impacts ischemic blood flow recovery. The increased blood flow recovery was observed in both whole-body JNK3 knockout mice and neural JNK3-deficient mice, but not in the muscle JNK3-deficient mice, demonstrating that neural JNK3 plays a key role in mediating blood flow recovery. Furthermore, we demonstrated that JNK3 activates a

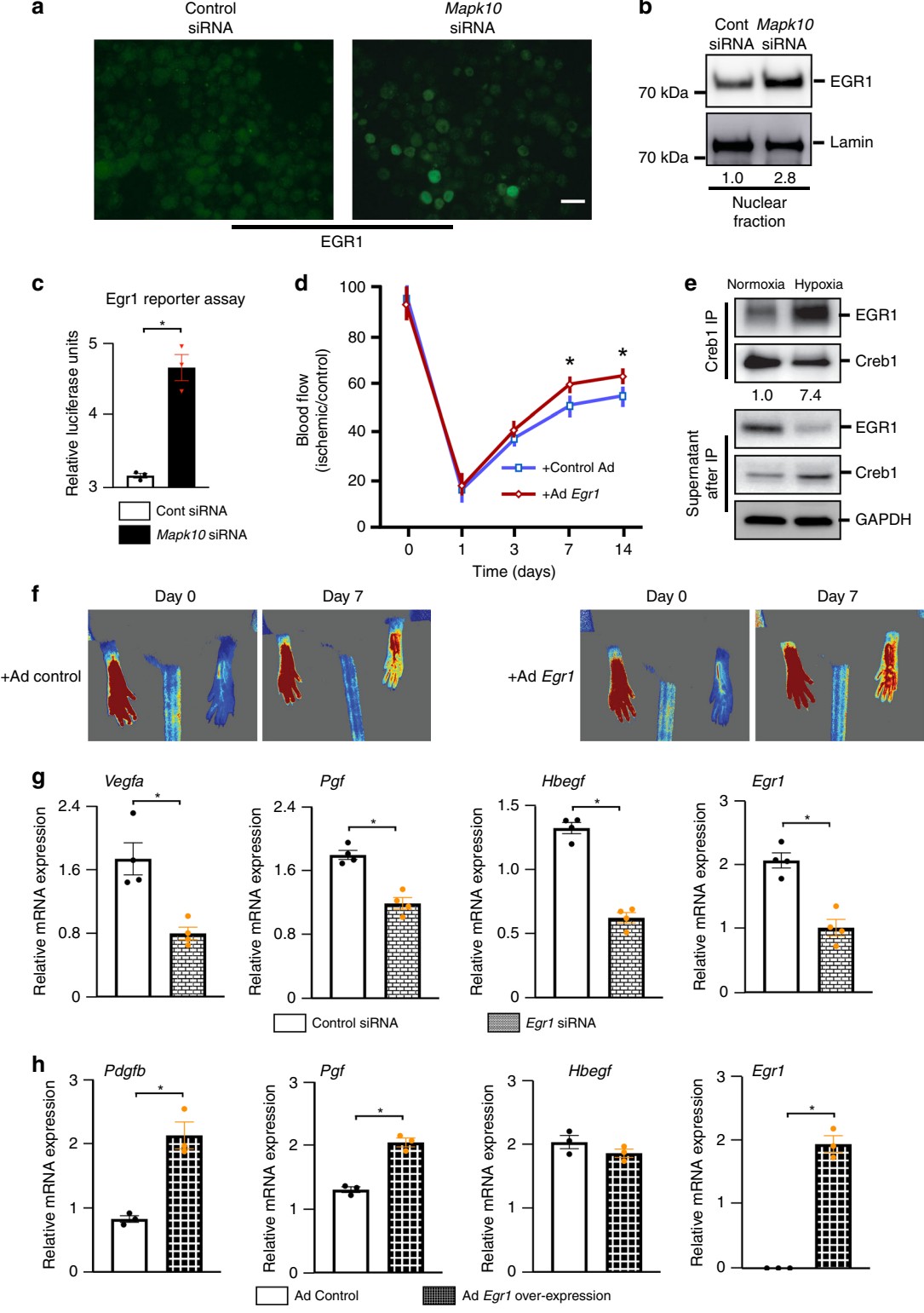

previously unknown JNK3–Foxo3a–Egr1/Creb1 signaling axis (Fig. 7h). Data presented in this manuscript clearly shows that JNK3 is required for Ser184/186 phosphorylation of 14-3-3 protein which, in turn, releases Foxo3a to the nucleus leading to suppression of the Egr1/Creb1 complex. In JNK3-deficient cells, Foxo3a no longer actively inhibits Egr1/Creb1, allowing these two transcription factors to work synergistically to activate transcription of genes involved in blood flow recovery. Thus, JNK3 deficiency improves blood flow recovery by allowing full activity of Egr1/Creb1 to promote neuronal growth factor expression that contributes to the revascularization response to ischemia.

JNK3 inhibition has been an active area of research for neurological disorders such as Parkinson's disease, Amyotrophic lateral sclerosis, and others[57,58]. However, neural JNK3 inhibition has not been previously recognized to play an active role in the vasculature, particularly regarding tissue ischemia. Other JNK family members, such as JNK1 and JNK2, have proven necessary for the developmental collateral artery formation. As a

**Fig. 5** Egr1 enhances the blood flow recovery after hindlimb ischemia. **a, b** Neuro-2a cells were treated for 48–72 h with scramble or *Mapk10* siRNA and exposed to hypoxia for 30 min. Then immunostaining was performed with an Egr1 antibody (**a**) and lysate was prepared after nuclear fraction isolation and immunoblotting was performed with Egr1 and control antibodies (**b**). **c** A Luciferase reporter assay for the Egr1 activity was performed on N2a cells treated with either control siRNA or *Mapk10* siRNA (48 h) and 90 min of hypoxia ($n = 3$ in each group). **d** Control Ad-GFP and Ad-*Egr1* were injected (single dose of $2 \times 10^8$ c.f.u.) into the gastrocnemius muscle of WT mice. Three days following the injection femoral artery ligation was performed and blood flow measurements by laser speckle contrast imaging (Control Ad, $n = 6$; Ad-Egr1, $n = 7$) were obtained. **e** Neuro-2a cells were treated with either normoxia or hypoxia followed by immunoprecipitation (IP) with a Creb1 antibody and immunoblotting with antibodies for Egr1 and Creb1. Supernatants were examined after IP by probing with antibodies as indicated. **f** Representative images of hindlimb blood flow of WT mice 7 days post femoral artery ligation and adenoviral (control Ad-GFP and Ad-*Egr1*) injection measured by laser speckle contrast imaging. **g** Following 90 min of hypoxia, RNA was isolated from control and *Egr1* knockdown N2a cells and qRT-PCR was performed for angiogenesis-related genes ($n = 4$ in each group). **h** RNA was isolated from control and *Egr1* overexpressing N2a cells and qRT-PCR was performed for growth factor related genes ($n = 4$ in each group). Statistically significant differences between groups are indicated (*$P < 0.05$ by Student's *t* test). The data are mean ± SEM. Scale bar, 20 μm. Source data are provided as a Source Data file

consequence, blood flow recovery from hindlimb ischemia is impaired in endothelial JNK1 and JNK2 compound knockout mice[15]. JNK3, in contrast, has only been examined in vascular tissue in a few studies and shown to be important in bovine endothelial cell proliferation[59] and retinal remodeling[60]. However, the work reported here expands on this literature and suggests that neural tissues are most relevant to the impact of JNK3 on the vasculature. Moreover, our work suggests neural JNK3 could be an important target for enhancing blood flow recovery in ischemic tissues. Indeed, our observations of neural-vascular cross-talk that impacts ischemic blood flow recovery suggests that strategies for therapeutic angiogenesis should not focus exclusively on vascular cells, but also target neural functions that support the vasculature.

The data presented here extends our knowledge concerning the impact of Creb1 in the peripheral nervous system. Creb1 is known to activate survival signaling in peripheral neurons[19,21], however, its role in ischemic blood flow recovery has not previously been described. We have shown that increasing the expression of Creb1, enhances growth factor expression in peripheral nerves and accelerates vascular repair after ischemia. Moreover, although it is known that the Creb1 co-activator, Creb binding protein (CBP) binds to Egr1[61], we have additionally shown that this Creb1/Egr1 interaction is enhanced by hypoxia. Similarly, Egr1 has previously been implicated in growth factor expression[62], however, its role in JNK signaling was not known. The data reported here indicates that JNK3 can now be included as the known modulators of Egr1. Thus, the data from this study indicates that neural Egr1 and Creb1 are important modulators of the vascular response to ischemia and mediate, in part, ischemic blood flow recovery in the hind limb. Given the ubiquitous distribution of peripheral nerves, one might speculate whether JNK3 inhibition, via its effects on Egr1/Creb, might impact other models of tissue ischemia or injury.

The JNK proteins can phosphorylate 14-3-3 protein at Ser184/186 which leads to release of Foxo3a from 14-3-3 binding[29]. In this paper we have clearly shown that 14-3-3 phosphorylation at Ser184/186 is dependent on JNK3 presence in neural cells. Furthermore, JNK3 appears to be required for Foxo3a to shuttle to the nucleus. In this regard, it has been reported that blood flow recovery is enhanced after hindlimb ischemia in Foxo3a knockout mice which the authors attributed this phenotype, in part, to enhanced eNOS[32]. The data reported here are generally consistent with those observations, but suggest that the lack of neural Foxo3a may also have contributed to the mechanism of enhanced blood flow recovery. Our data would predict that lack of neural Foxo3a would "uncouple" hypoxia-induced JNK3 activation from its inhibitory effect on growth factor gene expression. We expect that neural Foxo3a loss-of-function impacts ischemic blood flow recovery via promotion of Egr1 and Creb1 transcriptional activity and enhanced expression of growth factors. Thus, our data adds

neural JNK3 as a mechanism linking Foxo3a to the modulation of blood flow to ischemic tissues.

In summary, neural JNK3 deletion activates pathways involved in promoting blood flow recovery to ischemic limb tissue. As current treatments for peripheral vascular diseases in humans are limited in effectiveness, this JNK3–Foxo3a–Egr1/Creb1 pathway may serve as a promising target for therapies aiming to improve the peripheral vasculature in diabetic and other affected patient populations.

## Methods

**Human tissue biopsy.** All the human data have been acquired according to the all relevant ethical regulations. Human tissue biopsies were acquired with IRB exemption H00014081 in accordance with the guidelines of the Institutional Review Board of the University of Massachusetts Medical School. The tissue used was de-identified, without any patient information. We discussed our work with the Institutional Review Board at UMASS Medical School and the ethics committee waived the requirement for consent. In brief, tissue was harvested from amputated legs of patient undergoing leg amputation for critical limb ischemia. Tissue dissected at the proximal site where the limb was severed (where blood supply was deemed intact for wound healing), and also at a distal site several inches from the ischemic non-healing wound (where blood supply was compromised). Biopsies of nerve and muscle were sampled from these respective regions within 1 h of limb amputation in the University of Massachusetts Medical School Pathology lab.

**Animals and hindlimb ischemia model.** C57BL/6J strain mice were obtained from The Jackson Laboratories. Mice with global[63] and conditional *Mapk10*[64] gene deletion have been described previously. All the mouse experiments have been done according to all the relevant ethical regulations. Mice were housed in a facility accredited by the American Association for Laboratory Animal Care. All animal studies were approved by the Institutional Animal Care and Use Committee of the University of Massachusetts Medical School. Sarm1-null (#018069) mice were purchased from Jackson Laboratory and have been described previously[65]. Nestin cre (#003771, Nestin Cre targets precursor cells which includes neurons and Schwann cells) and HSA cre (*ACTA1* #006139) animals were obtained from Jackson Laboratory and crossed with conditional *Mapk10*-allele mice. Heterozygous *Mapk10*-null[63] animals on the C57Bl6/J background were bred to create homozygous $Mapk10^{-/-}$ and wild-type age-matched controls. Male mice at 8–12 weeks of age were anesthetized via intraperitoneal injection with a combination of 100 mg/kg ketamine hydrochloride and 5 mg/kg xylazine (Webster Veterinary, Devens, MA) before surgery. Unilateral hindlimb ischemia in the right leg was introduced in the mice[66]. In selected experiments, unilateral hindlimb ischemia was performed 3 days following injections of a single dose of 30 μL ($2.0 \times 10^8$ pfu), of either adenoviral vectors encoding Egr1 (Applied Biological Materials #087678 A), Creb1 (Vector Biolabs #1363), or GFP (a kind gift from Marcus Cooper, MD), into the gastrocnemius muscles. Hindlimb tissue perfusion was assessed with either moorLDI2-IR laser Doppler imaging system or moorFLPI-2 blood flow imager (Moor Instruments, Devon, UK). Blood flow images were obtained under conditions of constant body temperature (36 ± 1.0 °C) and average hindlimb blood flow was expressed as the ratio of ischemic to non-ischemic foot flow to account for minor variations in imaging conditions.

**Gene painting.** Gene painting in the right leg of mice was performed[51]. Mice at 10–16 weeks of age were anesthetized with intraperitoneal injection of 100 mg/kg ketamine hydrochloride and 5 mg/kg xylazine (Webster Veterinary, Devens, MA) or isoflurane before surgery. Surgically femoral and sciatic nerves were exposed in the right leg and painted using a small sterile brush with a solution containing 2 g/L poloxamer-F127 and $2 \times 10^8$ pfu/ml of the control GFP or shRNA for JNK3 adenovirus (Vector Biolabs #shADV-264201). The gene painting was followed by a

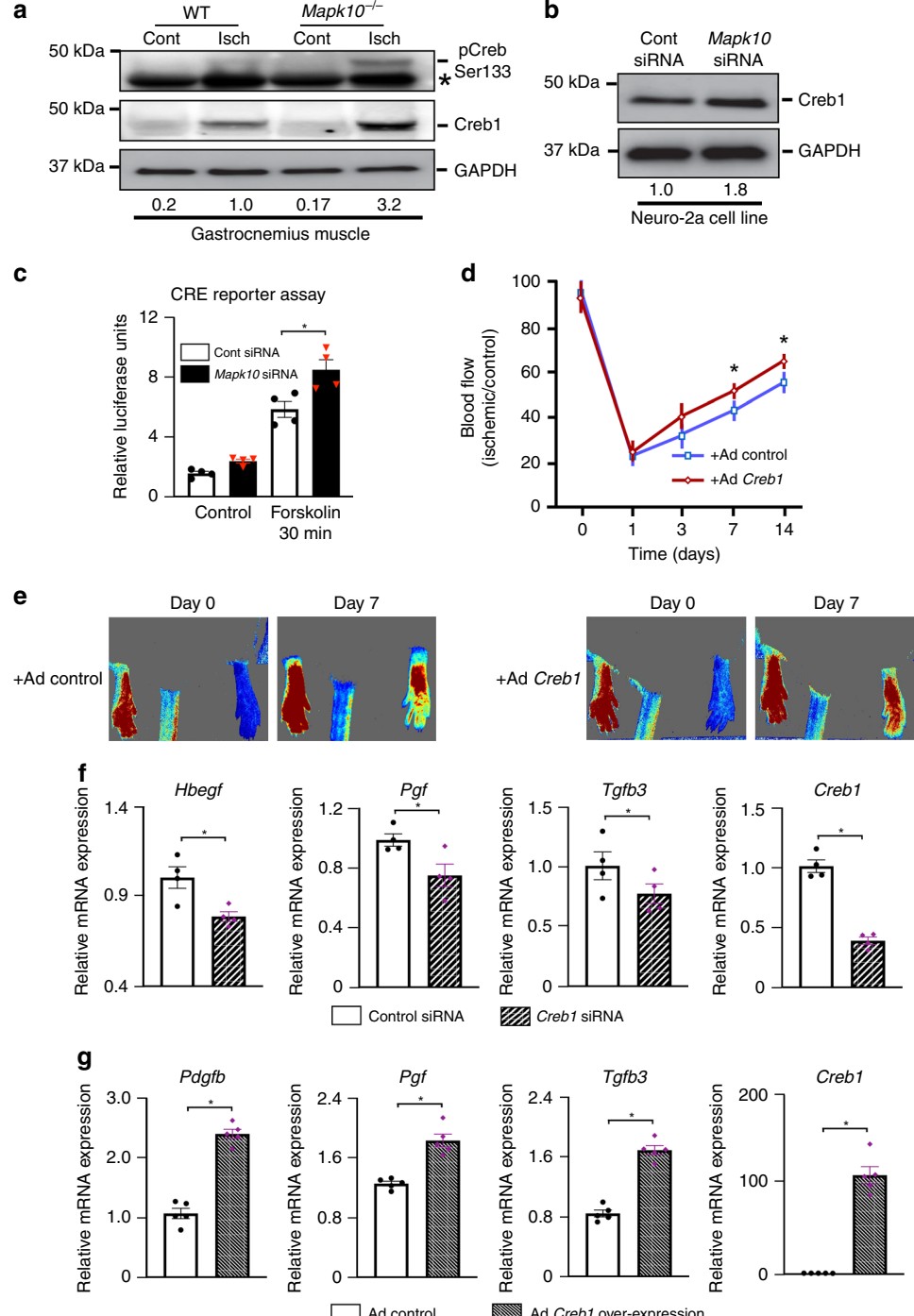

**Fig. 6** Creb1 transcription factor improves the blood flow recovery after hindlimb ischemia. **a** Gastrocnemius muscle was isolated from control and ischemic legs of both WT and *Mapk10*⁻/⁻ mice on day 3 after femoral artery ligation and immunoblot analysis was performed with antibodies for pCreb1 (phosphor-Ser133), Creb1, and GAPDH. **b** Lysates prepared from N2a cells treated with control siRNA or siRNA against *Mapk10* (48 h) were examined by immunoblot analysis using antibodies for Creb1 and GAPDH. **c** A Luciferase reporter assay for the Creb1 recombinase enzyme was performed on N2a cells treated with either control siRNA or *Mapk10* siRNA (48 h) both with and without Forskolin (30 min) a Creb1 activator (n = 4 in each group). **d** Control Ad-GFP and Ad-*Creb1* were injected (single dose of 2 × 10⁸ c.f.u.) into the gastrocnemius muscle of WT mice. Three days following the injection femoral artery ligation was performed and blood flow measurements by Laser Speckle Contrast Imager (n = 7 in each group) were obtained. **e** Representative images of hindlimb blood flow of WT mice 7 days post femoral artery ligation and adenoviral (control Ad-GFP and Ad-*Creb1*) injection were measured by laser speckle contrast imaging. **f** RNA was prepared from control and *Creb1* siRNA-treated neuro-2a cell line with hypoxia for 90 min and qRT-PCR was performed for genes indicated (n = 4 in each group). **g** RNA was isolated from control and *Creb1* overexpressing N2a cells and qRT-PCR was performed for growth factors related genes (n = 5 in each group). Statistically significant differences between groups are indicated (*P < 0.05 by Student's *t* test). The data are mean ± SEM. Source data are provided as a Source Data file

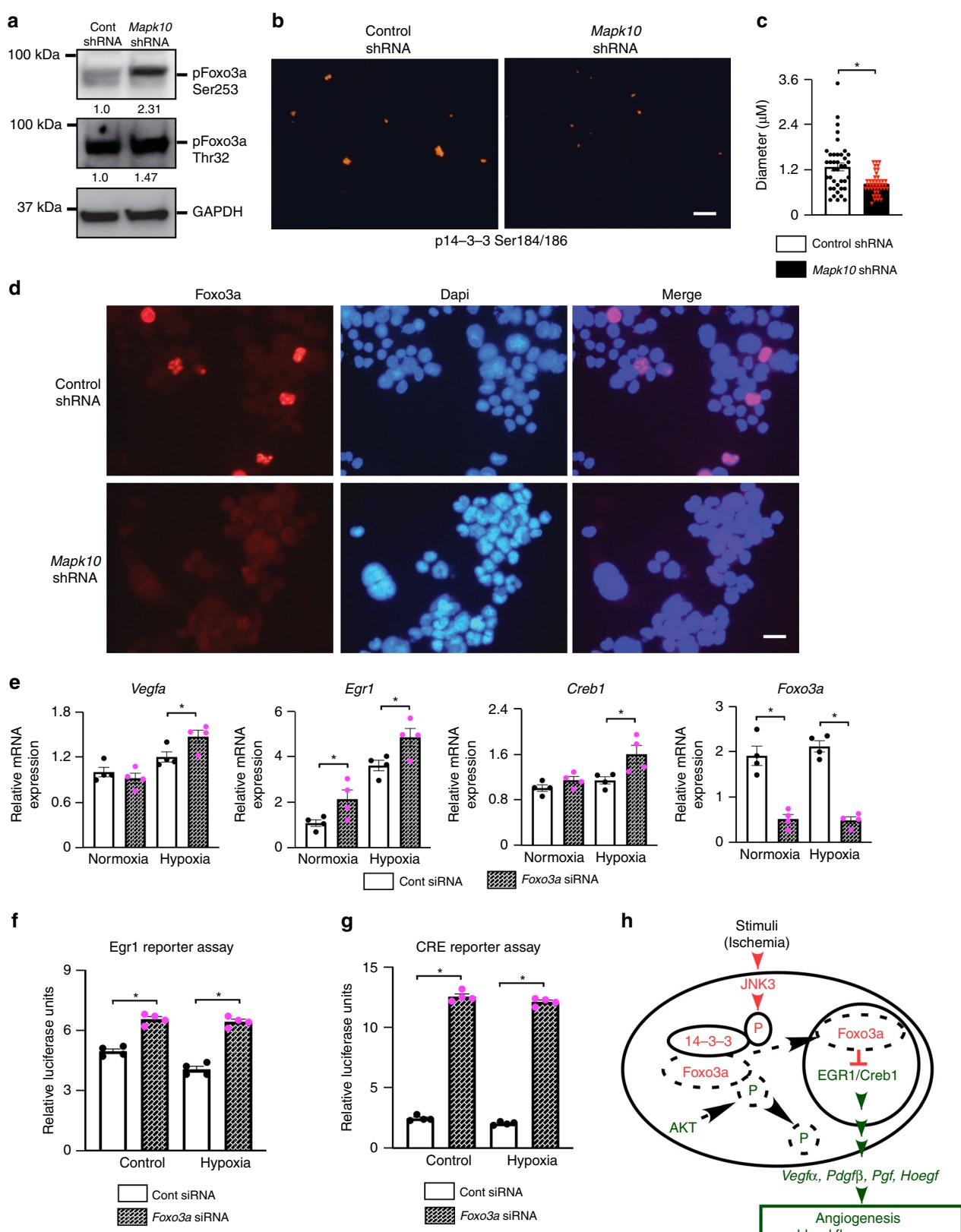

30 s rest period to allow for poloxamer drying. After the gene painting and brief drying period, the wound was closed. Post-surgery mice were given a recovery period up to a week to allow for reduced inflammation. Tissues were harvested and gene expression was analyzed by qRT-PCR.

**Cell culture and transfections**. Mus musculus brain neuroblastoma cell line Neuro-2a (#CCL-131) was purchased from ATCC and cultured in Dulbecco's

modified Eagle's medium supplemented with 10% fetal bovine serum, 100 units/ml penicillin, 100 μg/ml streptomycin, and 2 mM L-glutamine (Invitrogen).

Transfection assays were performed using 100 nM small interfering RNA oligonucleotides ON-TARGET plus SMART pool for control (D-001810-10), Jnk3 (Mapk10) (L-045023-00), Creb1 (L-040959-01), Egr1 (L-040286-00), and Foxo3 (L-040728-00) (Thermo Scientific Dharmacon, Lafayette, CO) in DharmaFECT 3 reagent (Thermo Scientific Dharmacon, T-2003) for 6–8 h in optimum

**Fig. 7** Foxo3a regulates the Egr1/Creb1 transcription factors activity and expression. **a**, **b** Neuro-2a cells were treated for 48–72 h with control or *Mapk10* shRNA, lysates were prepared, and immunoblotting was performed with pFoxo3a (Ser253), pFoxo3a (Thr32), and Gapdh antibodies (**a**) and immunostaining was performed with p14-3-3 beta/zeta (Ser184/186) antibody (**b**). **c** p14-3-3 beta/zeta (Ser184/186) quantification, in N2a cell line after treatment with control or *Mapk10* shRNA for 48–72 h (Control shRNA, $n = 39$; *Mapk10* shRNA, $n = 41$). **d** Immunostaining was performed with Foxo3a antibody in Neuro-2a cells after either treatment with control or *Mapk10* shRNA for 72 h. **e** RNA was isolated from control and Foxo3a siRNA-treated Neuro-2a cells and qRT-PCR was performed for genes as indicated ($n = 5$ in each group). **f**, **g** A Luciferase reporter assay for the Egr1 (**f**) and Creb1 (**g**) activity was performed on N2a cells treated with either control siRNA or Foxo3a siRNA (48 h) and 90 min of hypoxia ($n = 4$ in each group). **h** Schematic diagram of ischemia-induced JNK3 pathway during blood flow recovery. Statistically significant differences between groups are indicated (*$P < 0.05$ by Student's *t* test). The data are mean ± SEM. Scale bar, 5 µm (**b**) and 20 µm (**d**). Source data are provided as a Source Data file

(Invitrogen)[12]. The media was then changed to Dulbecco's modified Eagle's medium supplemented with 10% fetal bovine serum, 100 units/ml penicillin, 100 µg/ml streptomycin, and 2 mM L-glutamine (Invitrogen). After 48–72 h of siRNA treatment cells were exposed to low serum (0.5%) and hypoxia (1% oxygen) for 1 h in a hypoxia chamber (Billups-Rothenberg Inc.).

The cAMP response element (CRE) luciferase assay (Qiagen # CCS-002L) and early growth response 1 (EGR1) luciferase assay (Qiagen # CCS-8021L) were performed according to manufacturer's instructions.

**RNA preparation and quantitative polymerase chain reaction**. Total RNA was extracted from cells and tissues with the RNeasy Mini Kit (Qiagen) or TRIzol reagent (Invitrogen), and 1 µg of total RNA was reverse transcribed with oligo(dT) primers for cDNA synthesis using an iScript cDNA synthesis kit (Biorad). The expression of mRNA was examined by quantitative PCR analysis using a Quant-Studio™ 6 Flex Real-Time PCR System (Applied Biosystems). Taqman© assays were used to quantitate *Pdgfb* (Mm00440677_m1), *Tgfb3* (Mm00436960_m1), *Tfb1m* (Mm00524825_m1), *Tfam* (Mm00447485_m1), *Pgf* (Mm00435613_m1), *Npy* (Mm03048253_m1), *Npb* (Mm00462726_m1), *Mapk10* (Mm00436518_m1), *Hbegf* (Mm00439306_m1), *Creb1* (Mm00501607_m1), *Egr1* (Mm00656724_m1), *Vegfa* (Mm01281449_m1), *Foxo3* (Mm01185722_m1), *Tnfa* (Mm00443258_m1), *Cxcl1* (Mm00444662_m1), *Cxcl10* (Mm00445235_m1), *Cxcl2* (Mm00436450_m1), *Ccl2* (Mm00441242_m1), *Ucp3* (Mm00494077_m1), *Cd4* (Mm00442754_m1), *Cd8* (Mm01182108_m1), *Cd68* (Mm03047343_m1), *Ncam1* (Mm01149710_m1), *Hprt* (Mm00446968_m1), and *Gapdh* (4352339E-0904021) mRNA (Applied Biosystems). The $2^{-\Delta\Delta CT}$ method is used for relative quantification of gene[67,68]. Reference genes of *Hprt* and *Gapdh* were used to normalize the PCRs in each sample.

**Antibodies and immunoblot analysis**. Cell extracts were prepared using Triton lysis buffer [20 mM Tris (pH 7.4), 1% Triton X-100, 10% glycerol, 137 mM NaCl, 2 mM EDTA, 25 mM b-glycerophosphate, 1 mM sodium orthovanadate, 1 mM phenylmethylsulfonyl fluoride, and 10 µg/mL of aprotinin and leupeptin]. Protein extracts (50 µg of protein) in DTT-containing SDS sample buffer were separated in 10% or 12% SDS-polyacrylamide gels and transferred to Hybond ECL nitrocellu-lose membranes (GE Healthcare, Piscataway, NJ) and incubated with primary antibody with 1:1000 dilution. Immunocomplexes were detected with an Amer-sham™ Imager 600 using Immobilon Western HRP Substrate (EMD Millipore). Primary antibodies were obtained from Cell Signaling (phospho-Creb1 #9198, Creb1(rabbit) #9197, Creb1(mouse) #9104, Egr1 #4153, JNK3 #2305, Foxo3a #2497, Foxo1 #2880, phospho-Foxo3a (Ser318/321) #9465, phospho-Foxo3a (Thr32) #9464, phospho-Foxo3a (Ser253) #9466, JNK #9252, phospho-JNK #9251); Fitzgerald (p14-3-3 beta/zeta (Ser184/186) #70R-32590); Santa Cruz (PDGF-B #sc-7878, VEGFa #sc-7269); and Thermofisher Scientific (phospho-Foxo1 #PA5-38275).

Antibodies used as controls were obtained from Abcam (Gapdh #ab8245), Sigma (Actin #A2103), and Cell Signaling (Lamin #4777, alpha-Tubulin #3873).

**Immunoprecipitation**. Cell extracts were prepared using Triton lysis buffer [20 mM Tris (pH 7.4), 1% Triton X-100, 10% glycerol, 137 mM NaCl, 2 mM EDTA, 25 mM b-glycerophosphate, 1 mM sodium orthovanadate, 1 mM phe-nylmethylsulfonyl fluoride, and 10 µg/mL of aprotinin and leupeptin] and incu-bated (16 h at 4 °C) with 10 µg control non-immune rabbit IgG (Santa Cruz) or with 10 µg rabbit antibodies to Creb1 (Cell Signaling #9197). Immunocomplexes isolated using Protein G Sepharose were washed (five times) with lysis buffer.

**Immunofluorescence staining**. Gastrocnemius and thigh adductor muscle of mice were isolated and fixed in 10% formalin. Sectioning was done by University of Massachusetts Medical school Morphology Core Facility. Slides were permeabilized with 0.2% Triton X-100 in PBS, blocked in 10% goat serum for 1 h at room temperature, followed by incubation with either von Willebrand factor (VWF) (Dako, USA), CD31 antibody (BD Biosciences #553370), Foxo3a #2497(Cell Sig-naling) or Frtzgerald (p14-3-3 beta/zeta (Ser184/186) #70R-32590 (dilution 1:200) overnight at 4 °C, and detected with anti-rabbit Alexa 488 (Sigma, USA), anti-mouse Alexa 594 (Abcam) conjugated antibodies. 4′,6-diamidino-2-phenylindole

(DAPI, 1/100, Roche) was used to stain nuclei and images were acquired with Confocal (Carl Zeiss) and ZEN 2012 software (Carl Zeiss).

**Collateral vessel and capillary formation measurement**. Collateral vessels of the thigh adductor and capillaries gastrocnemius muscle were visualized via immu-nofluorescent staining for von Willebrand factor (VWF) or CD31. Capillary den-sity is expressed relative to the number of muscle fibers per high-power field area[39]. Muscle sections were imaged at ×20 magnification. Using ImageJ processing software (National Institute of Mental Health, Bethesda), fluorescent images were imported, and individual color channels separated. All area and intensity values were measured from the green channel. Under digital magnification, the maximal border of each hyperfluorescent vessel was manually outlined and the encompassed area measurement in pixels converted to µm$^2$ using the scale bar. From the cir-cumference of the vessel, diameter and area were calculated using the formulas for circle circumference and area, respectively. The capillary density of ischemic hin-dlimb muscle was directly compared with the contralateral sham non-ligated limb.

Quantification of fluorescence intensity was performed by means of a plug-in for ImageJ, named VesselJ. This macro is utilized in quantifying the sprouting of blood and lymphatic vessels in animal models given by choroidal neovascularization (CNV), though it is suggested for use in other models of noncorneal angiogenesis[69]. Total images per section were launched simultaneously for each analysis. The region of interest (ROI), to include autofluorescence, and background fluorescence intensity were manually defined. Pixels above the threshold value were counted as neovessels and pixels below the threshold were counted as non-vascularized pixels.

**Nuclear and cytoplasmic extraction**. Following the NE-PER Nuclear and Cyto-plasmic Extraction Kit (Thermofisher Scientific), cells were transfected with either control or *Jnk3* siRNA for 48 h and then either exposed to hypoxia or normoxia (15 min). After treatment cells were harvested with 0.05% Trypsin-EDTA (Ther-mofisher Scientific) and centrifuged at $500 \times g$ for 5 min. The resulting cell pellet was washed with PBS (1 ×) and the supernatant carefully discarded, leaving the cell pellet dry. Using the suggested provided reagents volumes depending on packed cell volume, reagent CER I and CER II were added to the cell pellet. Following a series of vortexing and an incubation on ice for 10 min, samples were micro-centrifuged at maximum speed ($\sim16,000 \times g$) for 5 min to obtain the supernatant (i.e., cytoplasmic extract). The produced pellet fraction was resuspended in NER reagent via incremental vortexing for a total of 40 min. Then samples were microcentrifuged at maximum speed ($\sim16,000 \times g$) for 10 min and the supernatant containing nuclear extract was immediately transferred to a new tube.

**Nerve injury by axotomy**. Using a disposable #15 scalpel blade, a 3–5 mm incision was made in the skin just below the top of the femur. A #7 dupont forcep was used to gently tunnel into the musculature to expose the nerve. The nerve was then resected from the femoral head to 0.5 cm above the knee using a small pair of spring scissors. All other structures (vein and artery) remained intact. A drop of bupivicaine (0.125%) was then applied to the injury site and the incision closed with 5.0 non-absorbable nylon suture and betadine applied. Sutures were removed 10–14 days later. Buprenorphine and ketoprofen were administered for analgesia as needed. Dissected nerve samples were prepared for plastic embedding and sec-tioning by the University of Massachusetts Medical School electron microscopy core facility.

**Statistical analysis**. All data are expressed as mean ± SE and the numbers of independent experiments are indicated. Statistical comparisons were conducted between two groups by use of Student *t* test or Mann–Whitney *U* test as appro-priate. Multiple groups were compared with either one-way Kruskal–Wallis or ANOVA with a post hoc Tukey–Kramer multiple comparisons test as indicated in legends. A *P* value < 0.05 was considered significant. All statistics were done using StatView version 5.0 (SAS Institute, Cary, NC) or GraphPad Prism version 5 (GraphPad Software, La Jolla, CA).

**Reporting summary**. Further information on research design is available in the Nature Research Reporting Summary linked to this article.

## Data availability

The data sets generated and analyzed as part of this study are available upon request from the corresponding authors. The source data underlying Figs. 1–7 and Supplementary Figs 1 and 7 are provided as a Source Data file. The gene expression data was deposited at NCBI's Gene Expression Omnibus (GEO). It is accessible through GEO series accession number of GSE135853.

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

## Acknowledgements

We thank Marta DeSourdis, Carol Marble and Donna Cobb for academic assistance and Xiaoyun Huang and Yongmei Pei for technical assistance and Anastassiia Vertii for critical reading. This work was supported by grants 16SDG29660007 from AHA (to S.K.), 5T32HL120823-03 (to K.V.T.), DK107220 (to R.J.D.), and HL092122, HL098407 (to J.F.K.) from NIH.

## Author contributions

S.K., S.M.C., K.C., K.R., K.V.T., R.J.D., O.P., M.F. and J.F.K. designed research. S.K., S.M.C., K.C., H.L., A.C., M.K., K.V.T., O.P. and M.R performed research. S.K., K.C., M.R, S.M.C., O.P., M.F. and J.F.K. analyzed the data. S.K., M.R., S.M.C., K.V.T., R.J.D. and J.F.K. wrote the paper.

## Additional information

**Competing interests:** The authors declare no competing interests.

**Peer Review Information** *Nature Communications* thanks Michael Courtney, Florian Limbourg and other anonymous reviewer(s) for their contribution to the peer review of this work. Peer reviewer reports are available.

