## [Peer Review File · Nature Communications]

Reviewers' comments:

Reviewer #1 (expert in hind limb ischemia-reperfusion animal models, arterogenesis and angiogenesis)

Remarks to the Author:

In the manuscript "JNK3 regulates angiogenesis via a Sirtuin1/Creb1 axis in peripheral tissue" Kant et al. Show JNK3 acts through Sirtuin1/Creb1 axis to coordinate vascular remodeling in response to ischemia. The authors employ a model of Hind Limb Ischemia to investigate the role of JNK3 in ischemia.

Although it is known that nerves express JNK3, the authors provide a novel role for JNK3 in vascular remodeling after ischemia and how JNK3 regulates pro angiogenic responses through the SIRT1/Creb1 axis. But the paper contains several major and minor concerns which preclude its publication in its current form.

Major concerns:

The authors of this study employ a general null allele of JNK3, but the effects of JNK3 deficiency are ascribed to peripheral nerve effects. This requires the study of a conditional allele.

Laser Doppler data are provided to demonstrate improved perfusion after hindlimb ischemia. The hindlimb ischemia model triggers arteriogenesis, which as a mechanism is capable to restore perfusion and involves remodelling of preexisting collaterals. This is fundamentally different to angiogenesis, which does not contribute to perfusion recovery. The effects on arteriogenesis, unfortunately, are not studied. Instead, capillary analysis is presented, which is triggered by ischemia.

Which C57Bl6 substrain was used for control and mutant mice? There are important differences in the N and J substrains. Genetic information should be provided.

A number of growth factors are upregulated in the Jkn3 null mice. Do the authors imply that this leads to enhanced perfusion recovery? Or are these observations true-true and unrelated? It is noteworthy that overexpression of some of these growth factors alone or in combination have failed in clinical trials.

Also, there seems to be a discrepancy in the quantitative data and the representative laser doppler images, which show color differences at day3, but not day 7. Unfortunately, no images are presented also for later timepoints. As a technical question, which region was analyzed for perfusion data?

Inflammatory cells like immune cells play a role in vascular remodeling by producing a variety of growth factors at the injury site. Is there a difference in immune cell reconstitution (monocytes, T cells, macrophages) between WT and JNK3-null mice?

The authors show that JNK3 is predominantly expressed by peripheral nerves at the ischemic site in comparison to endothelial cells. But the endothelial cells used for comparison are from the lungs which are not influenced by the ischemia (Fig. S1b). It is crucial to the manuscript that the authors can compare sorted endothelial cells with the peripheral nerves from the ischemic site for JNK3 expression.

The authors employ gene painting technique to introduce JNK3 siRNA into the right leg of mice. How specific is this technique? How do they ensure that they target only peripheral nerves? What is the impact of JNK3 siRNA on tissues other than nerves (endothelial cells, SMC). Without this data it is not possible to conclude a nerve specific pro angiogenic effect.

Do the JNK3-null mice present any phenotypic abnormality at baseline (control)? In Fig. 2g, VEGF A levels in JNK3-null mice at baseline (non ischemic) are increased compared the the WT mice at baseline. Are JNK3-null mice prone to pro-angiogenic responses even in the absence of ischemia?

The authors use neuro-2a extensively to elucidate their mechanism but the in vivo mechanism is supported by weak techniques which do not target the nerves specifically. How do the authors ensure nerve specific knockdown of JNK3 in their in vivo experiments.

How is the eventual increase of perfusion achieved in JNK3-null mice? In Fig. 1D from the representative images it looks like CD31+ cells in the JNK3-null mice at control conditions is less than the WT. Is a baseline difference in phenotype between WT and JNK3-null mice? What time point is being shown here?

Minor criticism:

1. The n numbers for most in vitro experiments are missing. The statistical analysis although mentioned should be mentioned alongside each experimental legend for better clarity.

2. It would also be helpful if the authors could mention experimental techniques in greater detail. For ex:gene painting, how was it performed, which region?

2. The legends for Fig. 2b are misplaced and need to be placed alongside the graph to avoid confusion.

Reviewer #2 (expert in angiogenesis and signaling pathways)

Remarks to the Author:

The study by Kant and coworkers is focused on the role of JNK3, which according to the authors is expressed in peripheral nerves but not in endothelial cells, in regulation of angiogenesis and blood flow recovery from mouse hind limb ischemia. The authors show that Jnk3 deficiency leads to increased production of pro-angiogenic factors in the ischemic muscle and cells by repression of the transcription factor Creb1 via Sirtuin 1 (SirT1). Creb1 is more active and upregulates pro-angiogenic factors in Jnk3-deficient mice as well as after silencing of Sirtuin1. They suggest that a JNK3/Sirtuin1/Creb1 axis coordinates vascular remodeling response to peripheral ischemia.

Major comments

1. The mouse model has apparently been described previously but no reference is given. It is assumed however that it is a constitutive knockout. Please provide information about the model, and if it is derived from another academic institution, it should be sequenced to verify its identity as a Jnk3 knockout. Please also verify that vascular development and vessel density is not affected in the naïve mice. From Fig. 1d it looks like vascular density may be lower in the knockout than in the wt? Please discuss and reference the paper by Salvucci et al. (Nat Commun. 2015 Mar 26;6:6576. doi: 10.1038/ncomms7576. EphrinB2 controls vessel pruning through STAT1-JNK3 signalling) which shows that Jnk3 indeed is expressed in certain vessels during development. The increased level of

VEGFA expression in the Jnk3^{-/-} knockout unchallenged condition also suggests that there would be differences in vascular parameters between wt and Jnk3^{-/-} mice.

2. It is difficult to see differences between the wt and Jnk3^{-/-} knockout in Fig. 1c at day 7 even though this is the time point when there is the biggest difference in flow in the ischemic muscle between the two. Are the panels in 1c representative? N=6 in each group – was the experiment repeated three individual times? In the legend to Fig. 1 and throughout, please give the number of observations for all the experiments. Quantify all blots, normalized to the total protein when examining phosphoproteins and to a house-keeping protein when applicable.

3. In Fig. 2, the authors show upregulation of pro-angiogenic factors in the Jnk3^{-/-} ischemic muscle. They compare the ischemic condition with the unchallenged one in the Jnk3^{-/-} mouse and conclude that there is no significant effect in expression of neuropeptides. However, at least for Npy and Nrf1 there is clear upregulation in either the basal or in the ischemic Jnk3^{-/-} condition compared to wt. Also Creb1 increases quite markedly during ischemia. In Fig. 2, please indicate significant changes between the ischemic condition in the wt and the Jnk3 knockout. It is unfortunate that the authors used pooled material for the array; it is much preferable to see the variation between individuals within and between the groups. In the b-g panels, please give n and number of repeats. Quantify the blots. Also please show the effect of Jnk3^{-/-} deletion on Vegfa transcripts (not only protein levels by blotting).

4. In Fig. 3, the authors again examine the levels of various genes after Jnk3 deletion. In a) they show expression in peripheral nerves. Please show enrichment of a neuronal marker to ensure that the correct tissue is examined. In b) verify specific knockdown of Jnk3 with the siRNA and not by the control siRNA. Compare expression of Jnk3 in Neuro2a and C2C12 cells or otherwise, panel c) is meaningless. Panel d shows no increased expression in protein levels of Creb1 although Figs 2 and 3 show increased transcript levels of Creb1 in the absence of Jnk3 expression. Please explain. In e) show expression of JNK3 protein after siRNA silencing.

5. In Fig. 4 a and b show expression of Creb1 in the adenovirus-transduced tissues (control and Ad-Creb1) over the time of the analysis. Please show whether there is an increased acetylation and phosphorylation of Creb1 in vivo in the Jnk3 KO mouse and whether there are changes in SirT1 phosphorylation in peripheral nerves under these conditions. Thus, although the in vitro data support the existence of a Jnk3-SirT1-Creb1 pathway in peripheral nerves, it needs to be demonstrated also in vivo.

Minor comment

The organization of the manuscript is suboptimal for a full paper.

We are pleased that the reviewers found our work and role of JNK3 in vascular remodeling novel and interesting. The reviewer's comments were helpful in strengthening our existing data and suggesting new experiments that strengthen our manuscript.

The revised manuscript entitled "Neural JNK3 regulates blood flow recovery after hindlimb ischemia in mice via a Egr1/Creb1 axis" is greatly improved and includes new data generated from two tissue-specific mouse models (floxed -JNK3 crossed either with Nes Cre or HSA Cre).

The new *in vivo* data strongly supports the conclusion that the JNK3 expression in neurons, and not in muscle, is required for suppression of blood flow recovery after hindlimb ischemia (HLI) in mice. Furthermore, we have recently uncovered the underlying molecular mechanisms that are responsible for improved recovery after HLI. We show here that JNK3 controls Foxo3a activity, which suppresses the activity of two transcription regulators Egr1 and Creb1. In addition, we also have new data that Egr1 and Creb1 bind together after hypoxia and when overexpressed *in vivo* they improve the recovery after hindlimb ischemia in mice.

Additionally, new and exciting data was obtained from human samples. We demonstrate that in human patients with peripheral artery disease JNK3 expression goes up in hypoxic areas in comparison to non-hypoxic areas of the same patients.

Again, we greatly appreciate the thoughtful comments of the reviewers. We believe that the suggested changes and new data have now moved the work to a level acceptable for publication in *Nature communication*.

Please find the point-by-point response to the reviewers' comments below.

Reviewer1

Q. The authors of this study employ a general null allele of JNK3, but the effects of JNK3 deficiency are ascribed to peripheral nerve effects. This requires the study of a conditional allele.

Response: We thank the reviewer for this helpful suggestion. We have now generated data from two different tissue-specific JNK3 conditional knockouts. One is a neuron specific (Nestin Cre) JNK3 deletion and the other is a muscle specific (HSA- Cre) JNK3 deletion (Rebuttal letter Fig. 1).

Fig.1 Neural JNK3 controls blood flow recovery after hind limb ischemia (HLI). Time course of blood flow recovery by Laser Speckle Contrast Imager of Neural specific (a) and muscle specific (b) JNK3 knockout mice (n=6-8 in each group). Statistically significant differences between groups are indicated (*, $P < 0.05$).

Q. Laser Doppler data are provided to demonstrate improved perfusion after hindlimb ischemia. The hindlimb ischemia model triggers arteriogenesis, which as a mechanism is capable to restore

perfusion and involves remodeling of preexisting collaterals. This is fundamentally different to angiogenesis, which does not contribute to perfusion recovery. The effects on arteriogenesis, unfortunately, are not studied. Instead, capillary analysis is presented, which is triggered by ischemia.

Response: Thank you for this comment. We now have included the data about the remodeling of preexisting collaterals in neuron specific JNK3 KO (Rebuttal letter Fig. 2). We have now presented the data about remodeling of preexisting collaterals by using diameter measurements of collaterals as well as the gene expression. Accordingly, we have changed the title as well.

Fig 2. (a) Expression of *von Willebrand factor* (vWF) in mouse thigh adductor (TA) muscle on Day 21 after ligation of femoral artery. (b) Quantification of the diameter of collateral vessels, vWF, in mouse thigh adductor (TA) muscle on Day 21 after ligation of femoral artery. (c) qRT-PCR was performed for different genes related to collateral artery remodeling on thigh adductor (TA) muscle from the control and ischemic legs of WT and *NES^{JNK3-/-}* mice 3 days post femoral artery ligation. All the experiments have been repeated 3-6 times. Statistically significant differences between groups are indicated (*, $P < 0.05$).

Q. Which C57Bl6 substrain was used for control and mutant mice? There are important differences in the N and J substrains. Genetic information should be provided.

Response: These are all C57/BL6J mice. We have included this information in the manuscripts material and method section as required.

Q. A number of growth factors are upregulated in the *Jkn3* null mice. Do the authors imply that this leads to enhanced perfusion recovery? Or are these observations true-true and unrelated? It is noteworthy that overexpression of some of these growth factors alone or in combination have failed in clinical trials.

Response: Overall, we believe the pathways outlined in the manuscript contribute to the phenotype. We understand that arteriogenesis and angiogenesis are complex processes and that singular therapeutic approaches to these phenomena have been proven naïve in clinical trials. The

goal of our work is to explore novel pathways that contribute to ischemic blood flow recovery that might be exploited therapeutically. Thus, one might draw from our work the implication that modification of neuronal pathways (e.g., *JNK3* inhibition) may hold therapeutic promise. The focus on EGR1/CREB1 is offered as a contributing explanation for our findings, not necessarily a therapeutic option.

Q. Also, there seems to be a discrepancy in the quantitative data and the representative laser doppler images, which show color differences at day3, but not day 7. Unfortunately, no images are presented also for later timepoints. As a technical question, which region was analyzed for perfusion data?

Response: We thank the reviewer for pointing this out. We have changed figures (Fig. 1d) to represent the quantitative data in the manuscript. All the perfusion data is from lower limb.

Q. Inflammatory cells like immune cells play a role in vascular remodeling by producing a variety of growth factors at the injury site. Is there a difference in immune cell reconstitution (monocytes, T cells, macrophages) between WT and *JNK3*-null mice?

Response: We thank the reviewer for pointing out this possibility. We have examined the expression of monocytes as well as T-cells markers after HLI in control and *Jnk3* -null hindlimb and find no substantial difference in these markers expression (Supplementary Fig. 5).

Q. The authors show that *JNK3* is predominantly expressed by peripheral nerves at the ischemic site in comparison to endothelial cells. But the endothelial cells used for comparison are from the lungs which are not influenced by the ischemia (Fig. S1b). It is crucial to the manuscript that the authors can compare sorted endothelial cells with the peripheral nerves from the ischemic site for *JNK3* expression.

Response: This is an excellent point. We have proceeded as suggested and examined murine and human skeletal muscle microvascular cells from gastrocnemius muscle and demonstrated that they do not express any *JNK3* (Supplementary Fig. 1c).

Q The authors employ gene painting technique to introduce *JNK3* siRNA into the right leg of mice. How specific is this technique? How do they ensure that they target only peripheral nerves? What is the impact of *JNK3* siRNA on tissues other than nerves (endothelial cells, SMC). Without this data it is not possible to conclude a nerve specific pro angiogenic effect.

Response: The gene painting technique is specific to those tissues that are exposed to the pluronic gel. However, given the comments of the reviewer, we have now utilized tissue specific deletion to fully address this concern (as outlined in comment #1 above). With regard to the specificity for genetic changes due to gene painting, we did harvest the nerve directly and the RT-PCR results pertain only to RNA harvested from the nerves specifically (Supplementary Fig. 7).

Q. Do the *JNK3*-null mice present any phenotypic abnormality at baseline (control)? In Fig. 2g, VEGF A levels in *JNK3*-null mice at baseline (non ischemic) are increased compared the WT mice at baseline. Are *JNK3*-null mice prone to pro-angiogenic responses even in the absence of ischemia?

Response: Yes, we see some baseline up regulation in a few genes in JNK3 knockout. But these up regulations are more robust in hypoxic condition. As in Fig. 3g VEGFa goes from 2.1 fold to 9.2 folds.

Q The authors use neuro-2a extensively to elucidate their mechanism but the in vivo mechanism is supported by weak techniques which do not target the nerves specifically. How do the authors ensure nerve specific knockdown of JNK3 in their in vivo experiments.

Response: We agree on this point and have subsequently done the requested genetic models.

Q. How is the eventual increase of perfusion achieved in JNK3-null mice? In Fig. 1D from the representative images it looks like CD31+ cells in the JNK3-null mice at control conditions is less than the WT. Is a baseline difference in phenotype between WT and JNK3-null mice? What time point is being shown here?

Response: We thank the reviewer for pointing out this oversight. We have changed the figures to representative figures.

Minor criticism:

1. The n numbers for most in vitro experiments are missing. The statistical analysis although mentioned should be mentioned alongside each experimental legend for better clarity.

We have now included the numbers and statistics in each figure legends as requested.

2. It would also be helpful if the authors could mention experimental techniques in greater detail. For ex: gene painting, how was it performed, which region?

We have extended our material and methods section to accommodate this issue.

2. The legends for Fig. 2b are misplaced and need to be placed alongside the graph to avoid confusion.

We thank the reviewer for finding this oversight. We have moved the legends to the right place.

Reviewer #2

Q 1. The mouse model has apparently been described previously but no reference is given. It is assumed however that it is a constitutive knockout. Please provide information about the model, and if it is derived from another academic institution, it should be sequenced to verify its identity as a Jnk3 knockout. Please also verify that vascular development and vessel density is not affected in the naïve mice. From Fig. 1d it looks like vascular density may be lower in the knockout than in the wt? Please discuss and reference the paper by Salvucci et al. (Nat Commun. 2015 Mar 26;6:6576. doi: 10.1038/ncomms7576. EphrinB2 controls vessel pruning through STAT1-JNK3 signalling) which shows that Jnk3 indeed is expressed in certain vessels during development. The increased level of VEGFA expression in the Jnk3^{-/-} knockout unchallenged condition also suggests that there would be differences in vascular parameters between wt and Jnk3^{-/-} mice.

Response: We thank the reviewer for pointing out this oversight. The JNK3 knockout mice

reference is now included in the manuscript (Reference number 58). Jnk3 expression has been reported in HUVECs [1] and BAECs [2]. We also have found a small expression of JNK in the human cell line HUVEC (Supplementary Fig. 1d). However, we are unable to see any expression in the two different types of primary endothelial cell isolated from mice (Supplementary Fig. 1c). Although we cannot rule out that JNK3 may express in vessels during development or in some mouse vascular beds, our new data demonstrate that neuronal specific JNK3 is important in this process as our *Nes-Jnk3* knockout mice phenocopy our JNK3 knockout mice in regard to ischemic blood flow recovery.

Q 2. It is difficult to see differences between the wt and Jnk3^{-/-} knockout in Fig. 1c at day 7 even though this is the time point when there is the biggest difference in flow in the ischemic muscle between the two. Are the panels in 1c representative? N=6 in each group – was the experiment repeated three individual times? In the legend to Fig. 1 and throughout, please give the number of observations for all the experiments. Quantify all blots, normalized to the total protein when examining phosphoproteins and to a house-keeping protein when applicable.

Response: We have changed the figures to clearly include the timepoint and N in each of the figures and immunoblot quantification has been included in the figures.

Q 3. In Fig. 2, the authors show upregulation of pro-angiogenic factors in the Jnk3^{-/-} ischemic muscle. They compare the ischemic condition with the unchallenged one in the Jnk3^{-/-} mouse and conclude that there is no significant effect in expression of neuropeptides. However, at least for Npy and Nrf1 there is clear upregulation in either the basal or in the ischemic Jnk3^{-/-} condition compared to wt. Also Creb1 increases quite markedly during ischemia. In Fig. 2, please indicate significant changes between the ischemic condition in the wt and the Jnk3 knockout. It is unfortunate that the authors used pooled material for the array; it is much preferable to see the variation between individuals within and between the groups. In the b-g panels, please give n and number of repeats. Quantify the blots. Also please show the effect of Jnk3^{-/-} deletion on Vegfa transcripts (not only protein levels by blotting).

Response: We appreciate the suggestions of the reviewer and have modified the figures as requested.

Q 4. In Fig. 3, the authors again examine the levels of various genes after Jnk3 deletion. In a) they show expression in peripheral nerves. Please show enrichment of a neuronal marker to ensure that the correct tissue is examined. In b) verify specific knockdown of Jnk3 with the siRNA and not by the control siRNA. Compare expression of Jnk3 in Neuro2a and C2C12 cells or otherwise, panel c) is meaningless. Panel d shows no increased expression in protein levels of Creb1 although Figs 2 and 3 show increased transcript levels of Creb1 in the absence of Jnk3 expression. Please explain. In e) show expression of JNK3 protein after siRNA silencing.

Response: We thank the reviewer for pointing out this oversight. Neural markers are now included in the manuscript (Supplementary Fig. 7). We have made the requested comparisons and these data are available in Supplementary Fig. 1b.

Q 5. In Fig. 4 a and b show expression of Creb1 in the adenovirus-transduced tissues (control and Ad-Creb1) over the time of the analysis. Please show whether there is an increased acetylation and phosphorylation of Creb1 in vivo in the Jnk3 KO mouse and whether there are changes in SirT1 phosphorylation in peripheral nerves under these conditions. Thus, although the in vitro data support the existence of a Jnk3- SirT1-Creb1 pathway in peripheral nerves, it needs to be

demonstrated also in vivo.

Response: Further investigation, as suggested by the reviewers, has brought to light a distinct mechanism for our observations that include CREB and EGR1. These are now the focus on the mechanistic experiments.

Minor comment□: The organization of the manuscript is suboptimal for a full paper.

We have rewritten the manuscript to be better organized.

1. Salvucci, O., et al., *EphrinB2 controls vessel pruning through STAT1-JNK3 signalling*. Nat Commun, 2015. **6**: p. 6576.
2. Pi, X., et al., *SDF-1alpha stimulates JNK3 activity via eNOS-dependent nitrosylation of MKP7 to enhance endothelial migration*. Proc Natl Acad Sci U S A, 2009. **106**(14): p. 5675-80.

Reviewers' comments:

Reviewer #1 (Remarks to the Author):

Reviewer 1

The authors provide new data generated with conditional alleles and tissue-specific Cre lines, which support their main conclusions.

There are still one experimental and some formal issues with the manuscript.

I would suggest to adopt an official nomenclature or style when referring to conditional alleles and tissue-specific Cre lines. E. g. HSA-Cre is termed HAS cre in the text (also typo), a reference should be given for both Cre strains.

The nomenclature for the generated conditional alleles is incorrect or misleading. E. g. the targeted, conditional alleles are termed NESJnk3^{-/-}, which is clearly wrong. I would suggest using either the proper genetic label (e. g. NesCre^{+/+} ; Jnk3^{f/f}) or a surrogate, such as Jnk3DNes

In general, the genetic nomenclature needs to be reviewed and adjusted to official usage, e. g. deletion of mouse Jnk3 allele should be indicated as Jnk3^{-/-} (not JNK3 deficient mice...).

Quantification of collateral artery diameter is provided, but no anatomical region is specified and the protocol/method and or reference is missing. Furthermore, since diameter varies depending on section angle or tissue compression, circumference measurements should be performed. See for example Limbourg, A et al, Nature Protocols, 2009.

The figure labels are now missing

Reviewer 2

The changes made to the manuscript should be indicated more specifically in the response letter. This would facilitate the judgement of the revised version.

I do not understand the comment to my previous question: Q 5. In Fig. 4 a and b show expression of Creb1 in the adenovirus-transduced tissues (control and Ad-Creb1) over the time of the analysis. Please show whether there is an increased acetylation and phosphorylation of Creb1 in vivo in the Jnk3 KO mouse and whether there are changes in SirT1 phosphorylation in peripheral nerves under these conditions. Thus, although the in vitro data support the existence of a Jnk3- SirT1-Creb1 pathway in peripheral nerves, it needs to be demonstrated also in vivo.

Response: Further investigation, as suggested by the reviewers, has brought to light a distinct mechanism for our observations that include CREB and EGR1. These are now the focus on the mechanistic experiments.

The authors suggest a new mechanism of Jnk3 deficiency via Foxo downregulation, which is interesting. However, this is in vitro only. While the abstract mentions this finding, is not clear that this is a suggested mechanism from in vitro experiments. Therefore, the abstract needs to be revised, and in vitro findings not supported by in vivo findings should be clearly indicated.

Reviewer #3 (Remarks to the Author):

The manuscript presents evidence for a specific role of JNK3 in neurons controlling the recovery of blood-flow after ischemia in the hind limb of mice. The evidence for the presence of neuronal JNK3 regulating blood-flow under these conditions appears strong. The authors find that changes in JNK3 levels in neurons (by either nestin-cre conditional knockouts or a tissue-specific RNAi method) or neuron-like cell lines (experiments in established cell lines) influences the levels, localization or phosphorylation of Egr, Creb and Foxo. These changes appear convincing. Together with the results of loss and gain of function experiments in cell lines, it leads the authors to a reasonable proposal that JNK3 acts on Foxo to regulate Egr and Creb, which releases specific pro-angiogenic factors when JNK3 is lowered.

The manuscript is clear and in general well presented. However I have two main comments that should be addressed as well as some minor points.

Major comments

1. The data presented (Fig. 1a,b) shows that levels of JNK3 protein and mRNA increase in ischemic tissue.

The activity of JNKs, as most protein kinases, are post-transcriptionally regulated by phosphorylation at specific residues in their activation loops. Although total levels of the kinase do influence overall activity, activation-loop phosphorylation is considered to be the usual mechanism of activation as, for MAPKs, activity is highly dependent on the state of phosphorylation (some 50000-fold increased turnover rate has been demonstrated in the case of ERK, PMID 11016942). The kinase activity of a kinase protein is not required for all its functions, but in the case of MAPKs this is expected.

Is there any direct evidence that there is an increase in activity of JNK3 under the conditions shown in Fig. 1a and b?

2. The presented link between JNK3 and Foxo is confusing and this should be improved.

First the introduction states JNK regulates Foxo transcription factors and cites papers reporting that DAF16 in *C.elegans* and sites in human Foxo4 transcriptional-activation domain are phosphorylated by JNK (in vitro). There is no mention that the Foxo4 JNK sites are not conserved in other Foxo family members, even though this is highly relevant to the manuscript (see below).

Experimental data is presented on Foxo3a even though it does not carry these TP sites that are the JNK substrates in Foxo4.

Data shown used an antibody specific to Foxo3 phosphorylation sites Ser319/322 (though this is not specified in the manuscript, only as "phospho-Foxo3a #9465").

These sites Ser319/322 are next to the nuclear export sequence and thought to regulate translocation not transcriptional activity like the JNK sites of Foxo4 referred to in the introduction, but this is not mentioned in the manuscript.

These sites Ser319/322 are reported to be substrates for casein kinases not JNK.

On the other hand, JNK has been reported to regulate the nuclear translocation of Foxo proteins. JNK can phosphorylate 14-3-3 proteins which provides a potential explanation for JNK action on Foxo localization.

All of these issues are clearly presented in a review (PMID 16288288).

The presentation of the Foxo aspects on the study should be reformulated to clarify these issues above.

Minor points

1. CREB phosphorylation is found to increase in JNK3 knockout (figure 6a). It is not clear if this should be expected, and what mechanism could be responsible. This deserves some comment.
2. Page 4 line 84-85: "required for neuronal survival". This should be corrected as ref 19 cited discusses how CREB is dispensable for survival of CNS neurons.
3. The methods section on human tissue should explain how quickly the tissue was harvested. Specifically, was the proximal site still oxygenated as it was at the time of amputation, was it ischaemic by the time samples were obtained and how was this determined? This may be relevant to main point 1, at least for fig. 1a.
4. There are a few typos in the text that need correction.
5. Figure parts, especially charts in the first few figures, often lack axis labels or they are incomplete
6. Figure 3a heatmap has colors (such as dark red) not in the scale bar
7. Figure 5e shows a dramatic loss of Egr1 from lysates after ischemia. Is this a consistent effect? If not, a more representative blot should be shown. If so, it appears to conflict with the positive regulation of Egr1 in the response and deserves some comment.
8. In the supplementary data what is mRNA relative to and why is JNK3 in C2C12 cells nearly 0.0 in figure S1b and 0.5 in figure S6.

The reviewers' enthusiasm for our work showing JNK3 as an important player in vascular remodeling after hindlimb ischemia was much appreciated. The insightful reviewer comments and suggestions resulted in experiments to further clarify the mechanism and resulted in a significantly strengthened manuscript.

In particular our manuscript has now been significantly improved using new human data and further elucidating the downstream pathways that control FoxO3a. Specifically, as requested by the reviewers, we have now included new human data clearly demonstrating JNK activation in the ischemic hindlimb (Supplementary Figs 1a and 1b). We have also included additional data demonstrating that JNK3 phosphorylates 14-3-3 to ultimately control Foxo3a function and nuclear cytoplasmic shuttling (Figure 7b, 7c and 7d).

Again, we greatly appreciate the helpful comments of the reviewers. Please find the point-by-point response to the reviewers' comments below.

Reviewer1

The authors provide new data generated with conditional alleles and tissuespecific Cre lines, which support their main conclusions.

There are still one experimental and some formal issues with the manuscript.

I would suggest to adopt an official nomenclature or style when referring to conditional alleles and tissuespecific Cre lines. E. g. HSACre is termed HAS cre in the text (also typo), a reference should be given for both Cre strains.

The nomenclature for the generated conditional alleles is incorrect or misleading. E. g. the targeted, conditional alleles are termed NESJnk3/, which is clearly wrong. I would suggest using either the proper genetic label (e. g. NesCre/+ ; Jnk3^{f/f}) or a surrogate, such as Jnk3DNes

In general, the genetic nomenclature needs to be reviewed and adjusted to official usage, e. g. deletion of mouse Jnk3 allele should be indicated as Jnk3^{-/-} (not JNK3 deficient mice...).

Response: We thank the reviewer for pointing this out. We have changed all the nomenclature accordingly (JNK3^{-/-} is *Mapk10*^{-/-}; Nes^{Jnk3^{-/-}} is Nes^{+Cre}; *Mapk10*^{f/f}). Also, we have corrected the typo in the text as well as provided the reference in text and methods.

Quantification of collateral artery diameter is provided, but no anatomical region is specified and the protocol/method and or reference is missing. Furthermore, since diameter varies depending on section angle or tissue compression, circumference measurements should be performed. See for example Limbourg, A et al, Nature Protocols, 2009.

Response: We thank the reviewer for this helpful suggestion. We now have provided information about the anatomical region of the collateral quantification. Furthermore, we now have included new data regarding the circumference measurement of collaterals in different genotypes (Supplementary Figure 3a and 3b).

The figure labels are now missing

Reviewer 2

The changes made to the manuscript should be indicated more specifically in the response letter. This would facilitate the judgement of the revised version.

Response: We thank the reviewer for pointing this out and have very specifically highlighted the changes in the manuscript and specified the figures in the letter. New figures included in the manuscript are Figure 7a, 7b, 7c, 7d Supplementary figure 1a, 1b, 3a and 3b. The changes to each section **have been highlighted in the text in yellow**.

I do not understand the comment to my previous question: Q 5. In Fig. 4 a and b show expression of Creb1 in the adenovirus-transduced tissues (control and Ad-Creb1) over the time of the analysis. Please show whether there is an increased acetylation and phosphorylation of Creb1 in vivo in the Jnk3 KO mouse and whether there are changes in SirT1 phosphorylation in peripheral nerves under these conditions. Thus, although the in vitro data support the existence of a Jnk3- SirT1-Creb1 pathway in peripheral nerves, it needs to be demonstrated also in vivo.

Response: Further investigation, as suggested by the reviewers, has brought to light a distinct mechanism for our observations that include CREB and EGR1. These are now the focus on the mechanistic experiments.

The authors suggest a new mechanism of Jnk3 deficiency via Foxo downregulation, which is interesting. However, this is in vitro only. While the abstract mentions this finding, is not clear that this is a suggested mechanism from in vitro experiments. Therefore, the abstract needs to be revised, and in vitro findings not supported by in vivo findings should be clearly indicated.

Response: We thank you for this suggestion. As requested, we have now clearly indicated that our Foxo3a data is derived from in vitro experiments. Although we have not provided *in vivo* data, the impact of Foxo3a gene deletion in the hindlimb ischemia model has been previously reported¹. The report indicates that Foxo3a gene deletion enhances ischemic blood flow recovery in the hindlimb ischemia model, as would be expected by our hypothesis. Thus, the reviewer's suggestions have allowed us to strengthened our hypothesis. As a consequence, we have included this information in the discussion section (please see highlighted portion of the discussion)

Reviewer3

The manuscript presents evidence for a specific role of JNK3 in neurons controlling the recovery of bloodflow after ischemia in the hind limb of mice. The evidence for the presence of neuronal JNK3 regulating bloodflow under these conditions appears strong. The authors find that changes in JNK3 levels in neurons (by either nestincre conditional knockouts or a tissuespecific RNAi method) or neuronlike cell lines (experiments in established cell lines) influences the levels, localization or phosphorylation of Egr, Creb and Foxo. These changes appear convincing. Together with the results of loss and gain of function experiments in cell lines, it leads the authors to a reasonable proposal that JNK3 acts on Foxo to regulate Egr and Creb, which releases specific proangiogenic factors when JNK3 is lowered.

The manuscript is clear and in general well presented. However I have two main comments that should be addressed as well as some minor points.

1. The data presented (Fig. 1a,b) shows that levels of JNK3 protein and mRNA increase in ischemic tissue.

The activity of JNKs, as most protein kinases, are posttranscriptionally regulated by phosphorylation at specific residues in their activation loops. Although total levels of the kinase do influence overall activity, activationloop phosphorylation is considered to be the usual mechanism of activation as, for MAPKs, activity is highly dependent on the state of phosphorylation (some 50000fold increased turnover rate has been demonstrated in the case of ERK, PMID 11016942). The kinase activity of a kinase protein is not required for all its functions, but in the case of MAPKs this is expected.

Is there any direct evidence that there is an increase in activity of JNK3 under the conditions shown in Fig. 1a and b?

Response: We thank the reviewer for this helpful suggestion. Unfortunately, there are no commercially available phospho-JNK3 specific antibodies, so we were not able to perform activation-specific immunoblots for JNK3. Moreover, despite occasional advertisements to the contrary, we have not reliably been able to perform immunoprecipitation with commercially-available JNK3 antibodies thereby preventing us from performing immune-complex kinase assays. As an alternative, we have now included experiments probing the phosphorylation state of all JNK family members (JNK1,2 and 3) in human muscle and nerve (Supplementary figure 1a and 1b). These data support the notion that JNKs are more active in ischemic tissue, particularly nerve.

2. The presented link between JNK3 and Foxo is confusing and this should be improved. First the introduction states JNK regulates Foxo transcription factors and cites papers reporting that DAF16 in C.elegans and sites in human Foxo4 transcriptional activation domain are phosphorylated by JNK (in vitro). There is no mention that the Foxo4 JNK sites are not conserved in other Foxo family members, even though this is highly relevant to the manuscript (see below). Experimental data is presented on Foxo3a even though it does not carry these TP sites that are the JNK substrates in Foxo4. Data shown used an antibody specific to Foxo3 phosphorylation sites Ser319/322 (though this is not specified in the manuscript, only as "phosphoFoxo3a #9465"). These sites Ser319/322 are next to the nuclear export sequence and thought to regulate translocation not transcriptional activity like the JNK sites of Foxo4 referred to in the introduction, but this is not mentioned in the manuscript. These sites Ser319/322 are reported to be substrates for casein kinases not JNK. On the other hand, JNK has been reported to regulate the nuclear translocation of Foxo proteins. JNK can phosphorylate 1433 proteins which provides a potential explanation for JNK action on Foxo localization. All of these issues are clearly presented in a review (PMID 16288288).

The presentation of the Foxo aspects on the study should be reformulated to clarify these issues above.

Response: We apologize for the error on our part. We also thank the reviewer for pointing us in a more fruitful direction. As suggested by the reviewer, we have now included new references in the manuscript describing that JNK controls the Foxo3a nuclear cytoplasmic shuttling via phosphorylation of 14-3-3 at Ser 184/186^{2,3}.

In addition, we are greatly indebted to the reviewer and have taken the reviewer's suggestion to investigate the possibility that JNK3 controls Foxo3a through regulation of 14-3-3. We have now generated new in vitro data that is consistent with a mechanism whereby JNK3 phosphorylates 14-3-3 on Ser184/186 thereby releasing Foxo3a to the nucleus where it inhibits Creb1 and Egr1 expression that contribute to pro-angiogenic growth factor expression. Thus, JNK3 acts as a restraint on growth factor expression. Inhibition of JNK3 releases this restraint. In support of this mechanism, shRNA-mediated JNK3 suppression reduces phosphorylation of 14-3-3 at Ser 184/186 (Fig. 7b), 7c such that Foxo3a remains in the cytosol and cannot translocate to the nucleus (Fig. 7d). In the absence of JNK3, this cytosolic Foxo3a is then subject to increased phosphorylation by AKT on Ser253 and Thr 32 (Fig. 7a). Thus, thanks to the reviewer we have been able to clarify the mechanism whereby JNK3 controls Creb/Egr1-mediated pro-angiogenesis gene transcription.

Minor points

1. CREB phosphorylation is found to increase in JNK3 knockout (figure 6a). It is not clear if this should be expected, and what mechanism could be responsible. This deserves some comment.

JNK3 activated Foxo3a can suppress the expression of Creb1 and therefore when JNK3 is absent, there is increased expression of total Creb1 protein (Figure 6a, b). Creb1 is phosphorylated and active in hypoxic conditions. Therefore, there is increased protein that is available for phosphorylation during hypoxia, hence we see enhanced Creb1 phosphorylation in *JNK3*-deficient cells compared to control. We have clarified this in the text

2. Page 4 line 8485: "required for neuronal survival". This should be corrected as ref 19 cited discusses how CREB is dispensable for survival of CNS neurons.

We thank the reviewer for finding the problem with this reference. This has now be corrected in the manuscript.

3. The methods section on human tissue should explain how quickly the tissue was harvested. Specifically, was the proximal site still oxygenated as it was at the time of amputation, was it ischaemic by the time samples were obtained and how was this determined? This may be relevant to main point 1, at least for fig. 1a.

We apologize for our lack of clarity and have now added further explanation of the tissue harvesting in the method section. All efforts were made to collect the tissue as soon as feasible to limit hypoxia.

4. There are a few typos in the text that need correction.

Thank you for this suggestion, we have now corrected several typos in the manuscript.

5. Figure parts, especially charts in the first few figures, often lack axis labels or they are incomplete

We apologize for this oversight. We have now corrected the axis labels.

6. Figure 3a heatmap has colors (such as dark red) not in the scale bar

Thank you for this suggestion, we have now put a new scale bar in the heatmap.

7. Figure 5e shows a dramatic loss of Egr1 from lysates after ischemia. Is this a consistent effect? If not, a more representative blot should be shown. If so, it appears to conflict with the positive regulation of Egr1 in the response and deserves some comment.

We apologize for the labeling. These data are supernatant which shows that the IP worked as EGR1 substantially binds to Creb1 after hypoxia and is reduced in the supernatant.

8. In the supplementary data what is mRNA relative to and why is JNK3 in C2C12 cells nearly 0.0 in figure S1b and 0.5 in figure S6.

Thank you for bringing this to our attention. We have now clarified the labeling in the manuscript.

1. Potente, M., *et al.* Involvement of Foxo transcription factors in angiogenesis and postnatal neovascularization. *J Clin Invest* **115**, 2382-2392 (2005).
2. Sunayama, J., Tsuruta, F., Masuyama, N. & Gotoh, Y. JNK antagonizes Akt-mediated survival signals by phosphorylating 14-3-3. *J Cell Biol* **170**, 295-304 (2005).
3. Greer, E.L. & Brunet, A. FOXO transcription factors at the interface between longevity and tumor suppression. *Oncogene* **24**, 7410-7425 (2005).

REVIEWERS' COMMENTS:

Reviewer #1 (Remarks to the Author):

My points have all been addressed or corrected.

One minor point: in the abstract, it should read: In JNK3 – deficient cell*s*, Foxo3a is suppressed....

Reviewer #3 (Remarks to the Author):

The authors have addressed a number of issues in their revision and clarified several points, thereby improving the manuscript. In my opinion this work represents an important advance in the understanding of the role of neural JNK isoforms in vascular remodelling, providing evidence for specific mechanisms by which JNK3 influences changes in the peripheral vasculature after ischemic injury.

I only have four minor suggestions

i) The labelling of figure 5e could be still further clarified by specifying "supernatant after IP" rather than just supernatant (since the lysates are themselves most likely "cleared supernatants")

ii) The figure 4 legend title refers to primary neuron and N2A but the data shows N2A cells and freshly isolated nerve and muscle, not primary neurons.

iii) Raw data for supplementary figures 1a and 1b could be included in the source data file

iii) A short comment regarding the neuronal specificity of the approach used would be very helpful. The present form of the manuscript refers at times to neural JNK3, and at others to neuronal JNK3 without addressing the distinction. Nestin Cre targets precursors of several cell types including neurons and Schwann cells (discussed e.g. in PMID 22372722), which are both present in peripheral nerves. A number of studies have proposed that JNK3 is expressed in Schwann cells as well as neurons (e.g. PMID: 15738268, 26339226).

Final revisions for manuscript NCOMMS-17-06794B

We are grateful to the reviewers for all their suggestions and comments resulting in a significantly strengthened manuscript showing our work on JNK3 as an important player in vascular remodeling after highlimb ischemia. As suggested by the reviewers we have made changes in the manuscript and data figures accordingly.

Again, we greatly appreciate the helpful comments of the reviewers. Please find the point-by-point response to the reviewers' comments below.

REVIEWERS' COMMENTS:

Reviewer #1 (Remarks to the Author):

My points have all been addressed or corrected.

One minor point: in the abstract, it should read: In JNK3 – deficient cell*s*, Foxo3a is suppressed....

Response: Thank you for bringing this oversight to our attention. We have made the proper correction.

Reviewer #3 (Remarks to the Author):

The authors have addressed a number of issues in their revision and clarified several points, thereby improving the manuscript. In my opinion this work represents an important advance in the understanding of the role of neural JNK isoforms in vascular remodelling, providing evidence for specific mechanisms by which JNK3 influences changes in the peripheral vasculature after ischemic injury.

I only have four minor suggestions

i) The labelling of figure 5e could be still further clarified by specifying "supernatant after IP" rather than just supernatant (since the lysates are themselves most likely "cleared supernatants")

Response: Thank you for this helpful suggestion. We have changed the labelling of this figure accordingly.

ii) The figure 4 legend title refers to primary neuron and N2A but the data shows N2A cells and freshly isolated nerve and muscle, not primary neurons.

Response: Changed accordingly as suggested by reviewers.

iii) Raw data for supplementary figures 1a and 1b could be included in the source data file

Response: Thank you for this suggestion. We have now included raw data for these supplementary figures in the source data file.

iii) A short comment regarding the neuronal specificity of the approach used would be very helpful. The present form of the manuscript refers at times to neural JNK3, and at others to

Final revisions for manuscript NCOMMS-17-06794B

neuronal JNK3 without addressing the distinction. Nestin Cre targets precursors of several cell types including neurons and Schwann cells (discussed e.g. in PMID 22372722), which are both present in peripheral nerves. A number of studies have proposed that JNK3 is expressed in Schwann cells as well as neurons (e.g. PMID: 15738268, 26339226).

Response: We have included a statement in our method section about the Nestin cre which deletes gene in neuronal as well as Schwann cells in Methods section.